# Ecogenomics reveals viral communities across the Challenger Deep oceanic trench

Ying-Li Zhou [1,6], Paraskevi Mara [2,6], Dean Vik[3], Virginia P. Edgcomb [2], Matthew B. Sullivan[3,4] & Yong Wang [1,5✉]

Despite the environmental challenges and nutrient scarcity, the geographically isolated Challenger Deep in Mariana trench, is considered a dynamic hotspot of microbial activity. Hadal viruses are the least explored microorganisms in Challenger Deep, while their taxonomic and functional diversity and ecological impact on deep-sea biogeochemistry are poorly described. Here, we collect 13 sediment cores from slope and bottom-axis sites across the Challenger Deep (down to ~11 kilometers depth), and identify 1,628 previously undescribed viral operational taxonomic units at species level. Community-wide analyses reveals 1,299 viral genera and distinct viral diversity across the trench, which is significantly higher at the bottom-axis vs. slope sites of the trench. 77% of these viral genera have not been previously identified in soils, deep-sea sediments and other oceanic settings. Key prokaryotes involved in hadal carbon and nitrogen cycling are predicted to be potential hosts infected by these viruses. The detected putative auxiliary metabolic genes suggest that viruses at Challenger Deep could modulate the carbohydrate and sulfur metabolisms of their potential hosts, and stabilize host's cell membranes under extreme hydrostatic pressures. Our results shed light on hadal viral metabolic capabilities, contribute to understanding deep sea ecology and on functional adaptions of hadal viruses for future research.

[1] Institute of Deep-Sea Science and Engineering, Chinese Academy of Sciences, Sanya, Hainan, China. [2] Department of Geology and Geophysics, Woods Hole Oceanographic Institution, Woods Hole, MA, USA. [3] Department of Microbiology and Center of Microbiome Science, The Ohio State University, Columbus, OH, USA. [4] Department of Civil, Environmental and Geodetic Engineering, The Ohio State University, Columbus, OH, USA. [5] Institute for Ocean Engineering, Shenzhen International Graduate School, Tsinghua University, Shenzhen, China. [6]These authors contributed equally: Ying-Li Zhou, Paraskevi Mara. ✉email: wangyong@sz.tsinghua.edu.cn

The global ocean is the largest virosphere on Earth and a reservoir of high viral diversity[1]. The role of viruses in the open ocean has been extensively described by the large-scale expeditions of Tara Oceans and Malaspina that revealed the high endemicity, structure, and lifestyle of epipelagic viral communities, as well as, a suite of adaptations that support their success[2–7]. Likewise, studies of viral metabolic reprograming of marine prokaryotes[8–10] suggested the potential for marine viruses to contribute to carbon and nutrient cycling in the ocean's water column by affecting the central metabolic pathways of their hosts.

Aside from the water column, viruses have also been identified in marine sediments, where they demonstrate extraordinary viral genetic diversity[11–16]. Nonetheless, viral communities in marine sediments are less studied than in water columns, due to the challenges of recovering viral particles efficiently from sediments[17–20]. Viruses show high abundances in marine sediments ($10^7$–$10^{10}$ particles $g^{-1}$ of dry sediment)[21]. Yet, the viral particles bind firmly to sediments due to electrostatic, van der Waals, and hydrophobic interactions, which complicate their separation and enumeration from the surrounding sediment matrix[21]. The challenges of efficiently separating viral particles from the sediments are due to the features of the virus (e.g., size, isoelectric point) and the sediment physiochemical properties (e.g., size, mineralogy, pH) that control the type and strength of interactions between viral and sediment particles[21–23].

Deep-sea sediments harbor ~160 Pg prokaryotic biomass[24], and in some cases, viral abundances in these settings are reported to exceed those of their putative prokaryotic hosts[13,14,25]. The viral shunt in abyssal and hadal realms is estimated to contribute 35% of labile carbon in those habitats and is believed to sustain the sediment microbiota in hadal sediments by providing easily degradable carbon[11,26]. Among prokaryotes, Thaumarchaeota and other archaeal lineages in deep-sea sediments, are reported to be more susceptible to viral infections compared to bacterial taxa[27].

The data on viruses that have been recovered so far from deep-sea sediments show extraordinary novelty, and can encode putative auxiliary metabolic genes (AMGs) involved in carbon and sulfur metabolisms[15,16,28–30]. These AMGs are suggested to enhance viral fitness and to impact the biogeochemistry of those habitats[15,31]. Recent studies of the New Britain trench identified novel viral clusters in sediments that have the potential to influence microbial hydrocarbon biodegradation at depths >8 km[32]. Still, studies of hadal viruses are limited[26,28,32,33] and have targeted only a few sampling sites, which further constrains our understanding of the biogeographic distribution, diversity, and genetic potential of viruses in these isolated hadal settings.

Here, we analyzed 37 sediment metagenomes and 3 metatranscriptomes for sediments collected from slope (>5 km) and bottom-axis sites (>10 km depth) along the Challenger Deep (CD) for the presence of viral elements. CD is the deepest hadal oceanic realm (~11 km depth) located at the southern end of the Mariana Trench, and is characterized by extreme hydrostatic pressures (>1000 atm), low temperatures (~2.5 °C), and a deficiency in labile nutrients[34,35]. The V-shaped topography of the trench creates a funneling effect that enhances the accumulation of organic carbon; CD bottom-axis sites present a twofold higher organic carbon content and sevenfold higher prokaryotic cell counts, compared to adjacent slope sites[35]. CD still remains one of the most oligotrophic hadal settings[36,37], which makes it challenging to explain these relatively high prokaryotic cell densities observed in bottom-axis sediments, and at the deeper sediment layers (>10 cm below sea floor; cmbsf)[35]. High viral production and turnover rates were reported in CD bottom-axis sediments[26], with viral density ranging between $2.4 \times 10^6$–$5.3 \times 10^7$ viruses $cm^{-3}$[28]. These numbers of viral particles could possibly provide labile organic carbon to sustain benthic prokaryotes in CD as has been described for other hadal trenches[26]. Whether the viral shunt and resulting prokaryotic turnover are linked to the high prokaryotic abundance in CD requires investigation of the lifestyle, metabolic potential, and virus-host interactions at different sites and sediment layers. The metagenomic analyses of viral communities collected from (hado) pelagic sediments in the northwest Pacific, including CD, showed that those viral communities were distinct from other marine habitats, with evidence for high endemicity[28,30]. Our previous study of microbial diversity in CD sediments revealed distinct prokaryotic communities between slope and bottom-axis sites[38,39], which leaves an open question of whether the spatial distribution of hosts influences the distribution of viruses in CD. Using in silico phage identification pipelines, we identified 1628 virus operational taxonomic units (vOTUs) within the 37 metagenomes and examined their taxonomy, viral community structure, and linkages to prokaryotic hosts. We also analyzed all viral contigs to identify putative AMGs that might provide additional insights into the roles of hadal sediment viruses. We present viral information from hadal metagenomes collected at different sites across CD that demonstrate distinct prokaryotic communities and geochemical gradients. Our study includes also viral data from the deepest region (>10,900 m) of this trench and describes the potential ecological implications of viruses in this extreme ecosystem.

## Results and discussion

We sequenced 37 microbial metagenomes from different depth horizons (2~3 cm intervals) of 13 sediment cores covering both slope and bottom-axis sites of the Challenger Deep (Fig. 1 and Supplementary Table 1). Clean reads of metagenomes from each site were co-assembled to yield 13 metagenomes, from which viral genomic data were extracted for analyses (Supplementary Data 1). We also generated three metatranscriptome libraries from one of the bottom-axis sediment cores (T3L11: 10,908 m; 6–9, 12–15, 18–21 cmbsf) to gain insights into potential viral activities.

**Identification and description of the CD viruses.** The assembled contigs from the CD metagenomes were analyzed initially using the What the Phage workflow, which utilizes output from 12 tools for phage annotation and identification[40]. We utilized 11/12 tools of this pipeline (Supplementary Data 2) and identified 9889 putative viral contigs with size >10 kb (https://doi.org/10.6084/m9.figshare.14815068). Due to the highly variable prediction quality of the tools in the utilized pipeline (Supplementary Data 2), we also performed manual and other curation approaches, to remove putative viral contigs that could be false positives (see also Methods). After strict and laborious curation, we retained 1628 contigs (1628/9889), which we dereplicated into vOTUs that represent species-level taxon ranks, using consensus metrics of >95% identity and >85% coverage[41–43]. Overall, 1622/1628 vOTUs were >10 kb while six had sizes <10 kb (Fig. 2a and Supplementary Data 3). The degree of completeness and contamination of the CD vOTUs was estimated by comparing the sequences using CheckV[44] against a large database of environmentally diverse and complete viral genomes. This resulted in assigning ~89% of identified CD vOTUs to four different quality tiers: complete genomes (73/1628 vOTUs; 100% completeness with direct terminal repeats), high- (79/1628; >90% completeness), medium- (193/1628; 50–90% completeness), and low-quality (1100/1628; <50% completeness) genomes (Fig. 2b). The completeness of 189 vOTUs, (~11%; 189/1628) could not be estimated (Fig. 2b). We also identified that 19% of vOTUs (316/

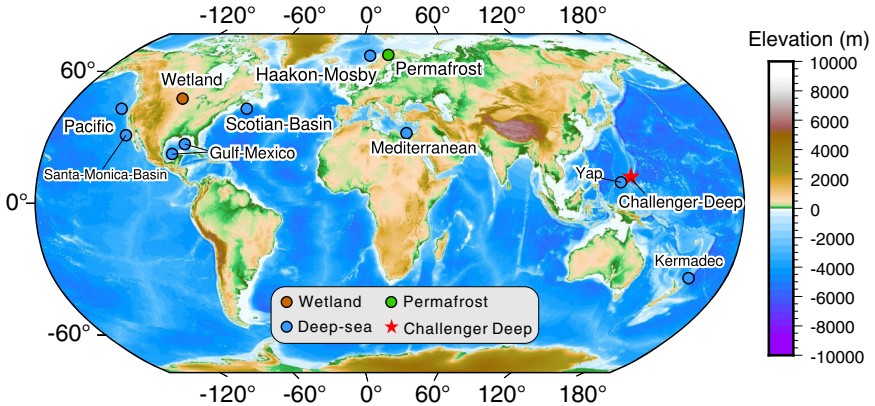

**Fig. 1 Global map indicating the Challenger Deep (this study; shown with a red star), the hadal and non-hadal deep-sea settings (blue), wetland (brown), and thawed permafrost (green) sites used for metagenomic comparisons in this study.** The map was generated by Yingli Zhou using the Generic Mapping Tools (v.6.0)[121] and SRTM15 + (V2.0) data[122].

1628) had at least 20% of genes mapped by >1 metatranscriptomic reads in our bottom-axis metatranscriptomic libraries (Supplementary Data 4 and 5).

To compare the CD vOTUs with those publicly available from other habitats, we used the gene sharing network analytic vConTACT2[45]. vConTACT2 clustered CD vOTUs at genus level with viruses deriving from pelagic seawater, sediment, and soil viruses (Fig. 1; Supplementary Fig. 1). We identified 1299 CD viral genera among the CD vOTUs. The majority of these genera (~77%; 1005/1299) were mainly distinct from the viral clusters deriving from pelagic seawater (Global Ocean Virome 2.0[7]), hadal and non-hadal deep-sea sediments (seven cold seeps[15] and three hadal trenches[30]), wetland sediments[46], and thawed permafrost[47] (Fig. 2c). The remaining ~23% CD genera overlapped with viruses from the hadal and non-hadal deep-sea sediments and seawater from the Global Ocean Virome 2.0[7] data sets, that were used for comparison (Fig. 2c and Supplementary Fig. 1). The distinct number of CD viral genera, and the limited overlap with other hadal and non-hadal deep-sea sediment habitats, indicate that these hadal CD viruses are presumably endemic to Challenger Deep.

Our CD vOTUs were also distinct when compared with viruses identified at the hadal slope sediments of the Mariana Trench[48]. Specifically, 98% of our CD viral contigs have not been previously identified in the Challenger Deep (<95% identity in 85% of sequence length). To be best of our knowledge, 76% of our identified CD viral genera were new (estimated by vConTACT2), when compared with the identified viruses from the upper slope (5.4–6.7 km depth) of the trench[48]. Comparisons of distinct viral populations between different settings in Challenger Deep (e.g., slope sites at various depths as well as slopes vs. bottom axis) will be beneficial for understanding hadal viral ecology and links between viral diversity and hadal physicochemical characteristics. However, unless additional locations are sampled at Challenger Deep in the future, the paucity of available comparisons limits the interpretation of hadal viral diversity in different settings.

We were able to assign taxonomy to 39% of the detected vOTUs using the majority-rules approach[7] (see Methods) (Fig. 2b). The CD vOTUs were mainly classified into three viral families that included Siphoviridae, Myoviridae, and Podoviridae. These viral families are well-classified in deep-sea sediments[16,30] and hadal water columns (Mariana, Yap, and Kermadec Trenches)[30] but also in pelagic settings (Tara Ocean)[7]. We note that since the time of data freeze for preparation of this manuscript, the taxonomy of phages has undergone a revision described in Walker et al. 2021[49] and is now implemented by the

International Committee on Taxonomy of Viruses (ICTV). As a result, the taxon naming will need to be updated by interested users of our data with the new taxon names that were approved after our analyses were completed. The estimated abundances of vOTUs were summed at the family level, and the taxonomically classified viruses accounted for 8% to ~54% of all viral communities (Fig. 3 and Supplementary Data 6).

Recruiting deep-sea metagenomic reads to our CD vOTUs showed that, 98% of our data were not detected in other deep-sea metagenomes used for comparisons in this study (Supplementary Fig. 2). This can suggest that the identified vOTUs in Challenger Deep are possibly endemic viral species of the CD trench. Among vOTUs, vOTU T1L10_NODE_10823 was the most abundant and accounted for ~4% (on average) of the viral communities across CD (Supplementary Data 5). vOTU T1L10_NODE_10823 shared homologous regions with other viruses in 2/15 deep-sea reference metagenomes (Supplementary Fig. 2). Highly endemic viruses have also been reported in the upper ocean, where local environmental conditions (e.g., oxygen, temperature) affect the host community structure[4,7]. Despite the increasing number of viral populations identified from various environmental settings[30,46,47], it appears that the deep-sea taxonomic diversity of viral communities still remains under sampled.

To evaluate the lifestyle of CD viral elements, we used VIBRANT[50] to predict prophage and integrase-encoding contigs as potential temperate viruses based on protein signatures (bacteria-like, and integrase-like genes) from KEGG, Pfam, and VOG databases[50]. Our results indicated that 1,541 viral contigs (95%) in CD viral communities were not assigned to either a lytic or lysogenic lifestyle (Fig. 2b, undetermined). It is possible that many/most of these undetermined viral contigs belong to viruses that have a lytic lifestyle in hadal depths. This would be consistent with studies of viral communities from surficial sediments collected in different deep-sea oceanic settings (Arctic, Atlantic, Pacific Oceans, and Mediterranean Sea; >1000 m water depth) that report high viral lysis rates[27]. With regard to lysogeny, it was predicted only in 5% of the CD viruses. This differs from deep-sea sediments that showed lysogeny as a more common potential viral lifestyle (e.g., Baltic Sea; ~19% on average)[25] but is more in line with the prediction results that we obtained for deep-sea cold seep sediments (7%)[15] and ocean seawater viruses (3%)[7] using VIBRANT[50]. Nonetheless, our arguments need to be interpreted with caution considering that 95% of viral contigs were not assigned as lytic or lysogenic.

To investigate the spatial distribution of the CD viruses, we estimated the relative abundance of CD vOTUs in the sediment

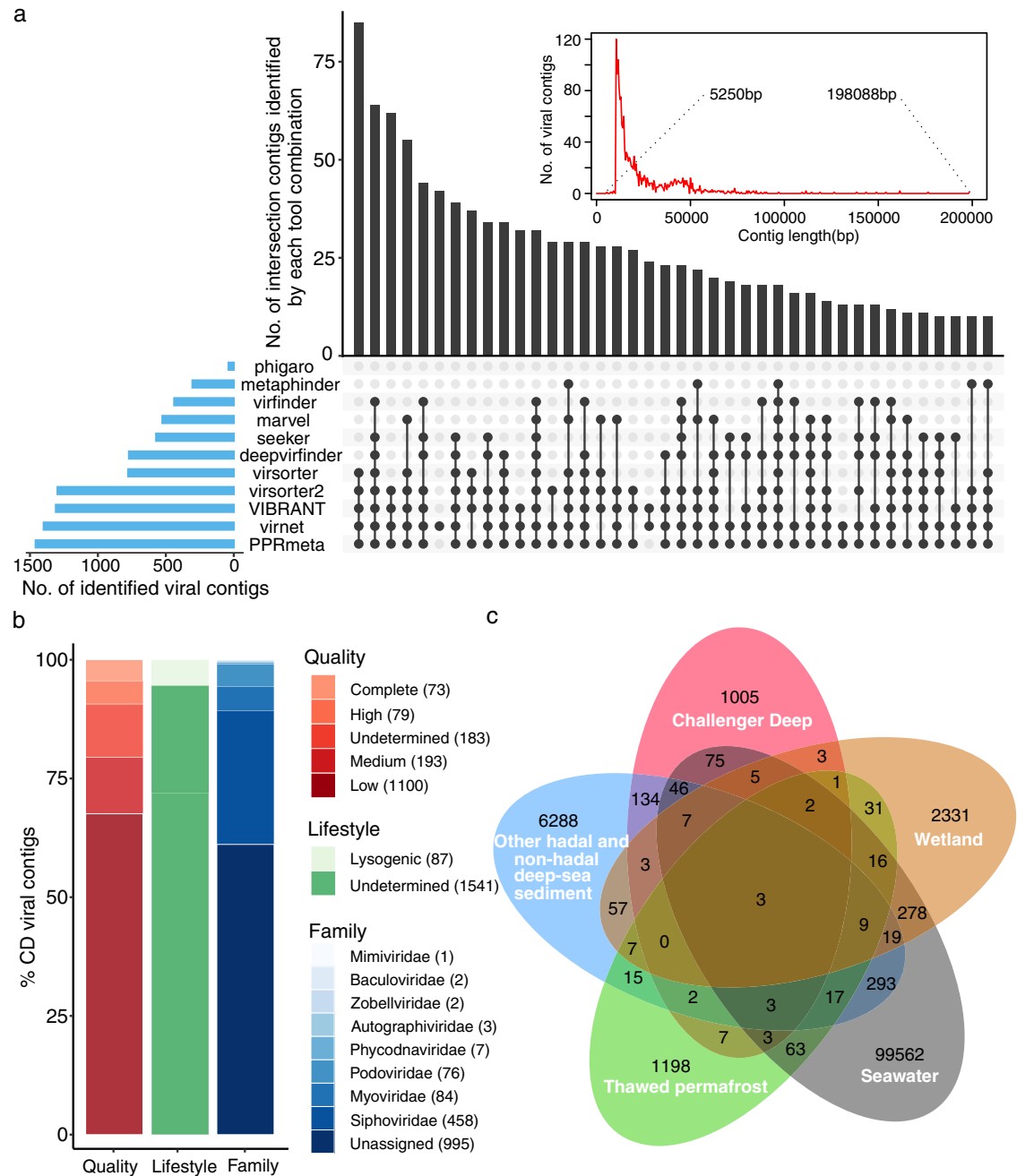

**Fig. 2 Overview of CD viruses and bona fide viral contigs (vOTUs) identified in this study. a** UpSet plot showing the vOTUs predicted from CD metagenomes by the 11 viral predication tools (bar chart on the left), the different combinations of multiple tools that predicted vOTUs (dot matrix on the bottom), the number of contigs identified by each tool combination (bar chart on the top), and the length distribution of all identified viral contigs (inset figure on the top). Dashed lines indicate the shortest and longest viral contigs, respectively. **b** Bar charts showing the quality and taxonomy of CD viral contigs. **c** Venn diagram of shared viral clusters (genus level) among the five data sets of environmental viruses from CD sediments (this study), other hadal and non-hadal deep-sea sediments (sediments from seven cold seeps[15] and three hadal trenches[30]), wetland[46], thawed permafrost[47], and pelagic seawater (Global Ocean Viromes 2.0)[7].

samples collected from slope and bottom-axis sampling sites across the trench. The relative abundance of vOTUs in each 2~3 cm sediment layer was calculated as the normalized coverage of each vOTU divided by the total normalized coverage of vOTUs at the investigated sediment layer (Supplementary Data 5). Principal coordinate analysis (PCoA) utilizing a Bray–Curtis dissimilarity distance matrix showed a significant difference ($p = 0.001$) between the vOTUs isolated from the slope vs. bottom-axis samples (Fig. 4a and Supplementary Fig. 2). The distribution of the dominant viral populations, at species level,

was also different between the slope and bottom-axis sites (Supplementary Fig. 3). This can be attributed to differences in the geographical isolation and nutrient availability between slope and bottom-axis sites that have been suggested to affect the distribution of prokaryotic communities across the V-shaped CD trench[38,39]. Differences in viral communities at discrete depths observed in this study and between this study and upper CD slope[48] may possibly reflect in situ variations in available nutrients, and/or variations in DNA recovery or methods used for metagenome assembly and extraction of viral data. In our

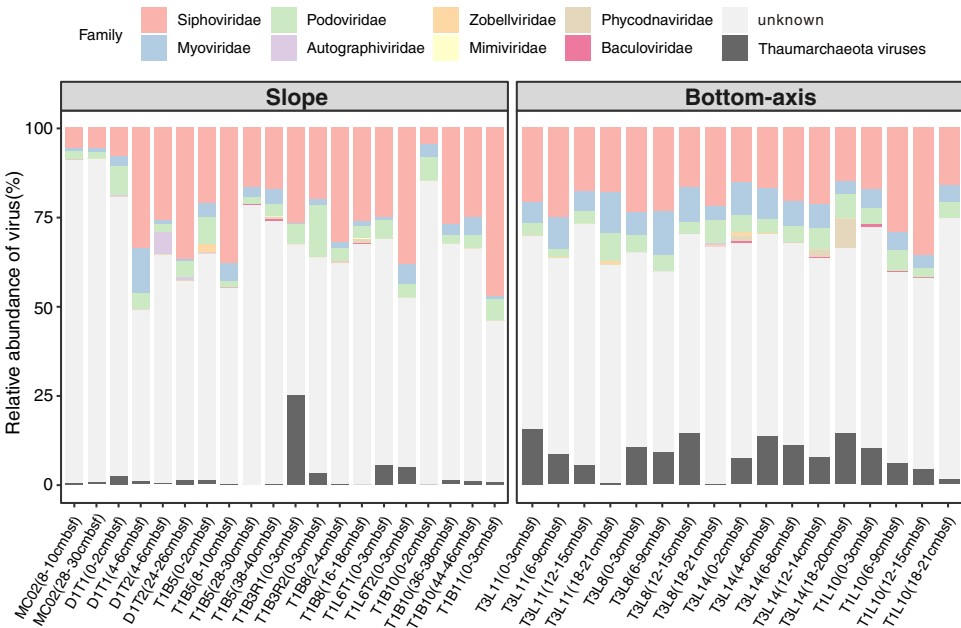

**Fig. 3 Relative abundance of viruses (family level) in 37 metagenomes.** The relative abundance of vOTUs in each sediment layer was calculated as the normalized coverage of each vOTU divided by the total normalized coverage of the vOTUs in each metagenome. The estimated abundances of vOTUs are summed at family level except for four potential Thaumarchaeota viruses, which are also unclassified.

study, there were also ubiquitous vOTUs such as T1B5_NODE_690, T1B8_NODE_8617, and T1B5_NODE_8075 that were present in all of the 37 CD samples, but could not be classified. The alpha diversity of CD viral communities was significantly higher ($p < 0.05$) in the topographically isolated bottom-axis, when compared to slope sites. The different diversity scores between the bottom-axis and slope were supported by all three indices (Chao1, ACE, and Shannon), as well as by the identified vOTUs that were overall discrete between bottom-axis and slope sites (Fig. 4b–f and Supplementary Fig. 4). Our results indicate higher viral community diversity, and distinct viral components in the bottom-axis sediments which are at deeper and more remote water depths, compared to the slope sites of the trench. Also, the bottom-axis sediments accumulate higher amounts of detrital organic matter due to the V-shaped topography of the trench, which could increase the role of organic matter in shaping microbial host communities and subsequently, viral diversity.

**Host and virus linkages**. The ecological role of CD viruses and their potential to affect nutrient cycling[5,15] across the trench was examined by screening 586 CD microbial metagenome-assembled genomes (MAGs) to identify putative hosts (NCBI BioProject accession: PRJNA635214). These prokaryotic MAGs were recovered from the same metagenomes as the viral contigs. For host prediction, we used VirMatcher[51], which is the only current host prediction tool to assign confidence scores (see Methods). We predicted potential prokaryotic hosts for 14 CD vOTUs (Supplementary Data 7), which accounted only for a small fraction (14/1628) of the CD viral community. The in silico host prediction indicated that CD viruses may infect 42 of our CD MAGs, assigned at 27 taxa at the species level (spanning seven phyla). These taxa include heterotrophs (e.g., Proteobacteria) and chemoautotrophs (e.g., Thaumarchaeota, Planctomycetota) involved in nitrogen and carbon cycling whose taxonomic signatures were abundant in CD sediments[39], but with different relative abundances (7% to 43%) across the discrete sampling sites (bottom-axis vs. slope) (Supplementary Data 7). Indeed,

Proteobacteria, Thaumarchaeota, and Planctomycetota were the most frequently predicted hosts in CD sediments (Fig. 5). Thaumarchaeota has been identified as potential hosts for archaeal viruses in deep-sea sediments collected from various oceanic realms[27]. In this study, Thaumarchaeota were identified as potential hosts of the most abundant vOTU (T1L10_NODE_10823), which was detected in 32/37 CD metagenomes and accounted for 7%~15% of viral community composition at aerobic top sediment layers (0–3 cmbsf) of bottom-axis sites (Supplementary Data 5).

Interestingly, we also identified four potentially new Thaumarchaeota viruses that were present in the CD viruses (Supplementary Data 7). These were mainly detected in top sediment layers of cores from four bottom-axis sites, while it comprised up to 25% of the viral community in sediment layers from one slope site (Fig. 3). We used vConTACT2 to cluster these previously undescribed Thaumarchaeota viruses with 119 other known marine archaeal viruses[10,52,53]. Our four identified Thaumarchaeota viruses were distinct, and did not cluster with other known archaeal viruses. This indicates that these viruses are possibly hadal Thaumarchaeota-related viruses endemic to Challenger Deep. However, this requires further investigation.

VIBRANT predicted that chemoautotrophic taxa involved in nitrogen cyclings such as *Scalindua* (Planctomycetota), *Nitrospinaceae* (Nitrospinota), and *Nitrososphaerales* (Thaumarchaeota) would be infected by lytic viruses in our CD sediment samples (Supplementary Data 7). Nonetheless, recent studies from coastal waters have reported viral isolates (e.g., *Nitrosopumilus*; spindle-shaped viruses) from marine Thaumarchaeota that cause chronic infections accompanied by growth inhibition of the host and severe reduction in rates of ammonia/nitrite oxidation/reduction[10]. Based on our analyses, lysogeny is a less likely lifestyle (5% assigned) in our identified CD viral contigs. Yet, the inability to assign lifestyle to the majority of the viral contigs (95%) might underestimate the importance of lysogeny, while at the same time preventing us from predicting the lytic viruses in CD. We suggest that lytic infections (if occurring) might be important and affect available nutrient pools across the V-shaped

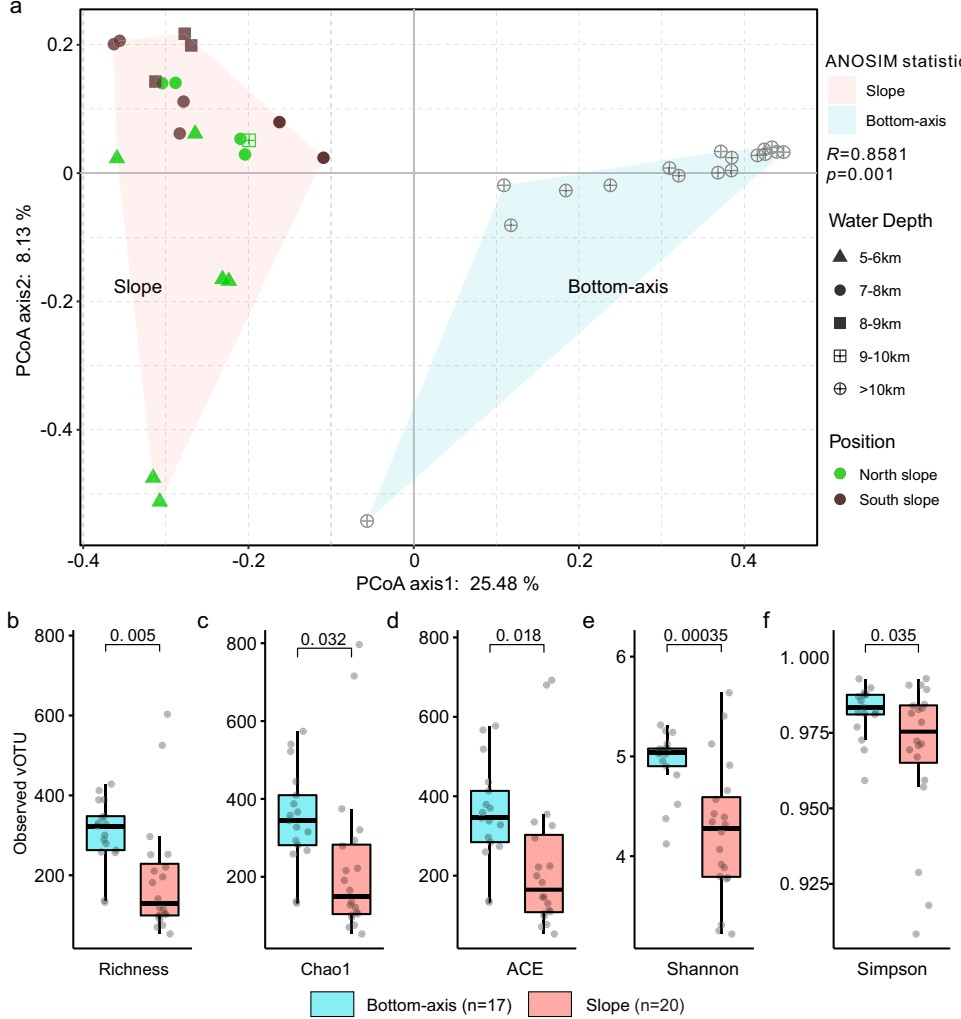

**Fig. 4 Alpha- and beta-diversity of CD viral communities. a** Bray–Curtis dissimilarity principal coordinate analyses (PCoA) of the viral communities. The pink and blue areas covered slope and bottom-axis samples, respectively. **b–f** Estimated number of vOTUs, Chao1, ACE, Shannon, and inverse Simpson indices of the viral community diversity from the CD slope (red, $n = 20$) and bottom-axis (blue, $n = 17$) sediments. $p$ value was estimated using the two-sided Student's $T$ Test. For boxplots, center line indicates median, bounds of box indicate 25th and 75th percentiles, and whiskers indicate minimum and maximum.

Challenger Deep (bottom-axis vs. slopes sites). This potential virus-induced effect on nutrient availability could act as a selective force in shaping microbial composition across the CD[36,37].

Many of the chemoautotrophs reported to be susceptible hosts of lytic viruses at CD can also affect carbon pools due to their activities as carbon fixers. Nitrospinota as well as Nitrososphaerales are important carbon fixers in the dark ocean[54], and are among the predicted CD hosts. Similarly, we suggest that CD viruses (if indeed lytic) affect pools of available labile organic carbon along the CD by affecting host populations that transform nitrogen pools and fix carbon. This can shape the distribution patterns of prokaryotes and associated viruses in CD sediments as suggested for other deep-sea sediments[15,27] (Fig. 4a). Our arguments require further experimental and culture-based investigations; however, viruses are known to regulate energy gain processes that occur in the deep subsurface biosphere, and recycle and/or divert the flow of carbon in the global ocean when they destroy or manipulate their hosts[55,56].

The predicted potential prokaryotic hosts for the 14 vOTUs may suggest that CD viruses target specific prokaryotic hosts in these CD sediments; however, this requires cautious interpretation considering that our host predictions were successful for ~1% of the viral population that we identified. We detected only a small fraction of predicted hosts (16%) that could be possibly infected by more than one vOTU, while only six vOTUs had multiple potential hosts at the species level. One vOTU (T1B5_NODE_7184) had the potential to infect different *Rhodospirillales* taxa (Supplementary Data 7), which can be abundant in surficial Mariana Trench sediments[57].

**Putative metabolic genes in CD viruses.** To further explore the potential ecological role of identified viral elements in CD sediments, we examined the VIBRANT and DRAM-v annotations of CD vOTUs (Supplementary Data 8) for putative viral metabolic genes. Putative viral metabolic genes, like AMGs, can affect the efficiency of host-microbial metabolic pathways[15,47,58] that often encode central metabolic enzymes[59]. We searched for candidate AMGs using the DRAM-v pipeline, and performed manual curation to verify their viral origin and position on the viral contigs (see Methods). We were able to identify 249 putative AMGs (Supplementary Data 8), most of which are affiliated with amino acid and carbohydrate metabolisms, and the production of

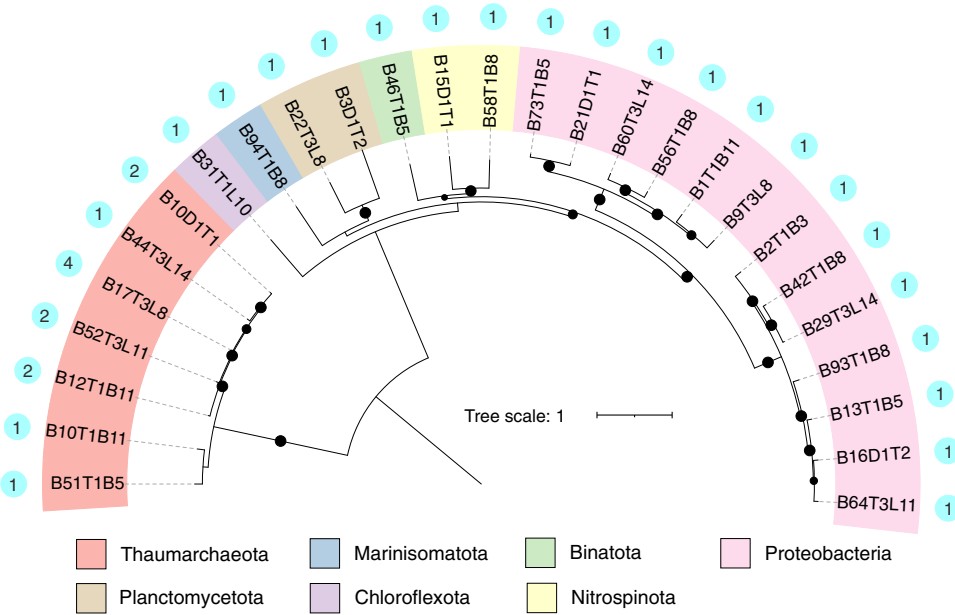

**Fig. 5 Challenger Deep virus-host linkages.** Maximum-likelihood phylogenetic tree of prokaryotic genomes (phylum level). The outermost dots denote the number of viruses that infected the species.

cofactors and vitamins (Supplementary Fig. 5 and Supplementary Data 9). We compared the putative AMGs from the bottom-axis and slope sites for potential differences in abundances and metabolic functions. Overall, no discrete separation between the putative AMGs from the different sampling sites was observed despite the apparent topographical separation of the viral communities (bottom-axis vs. slope sites; Fig. 4a and Supplementary Fig. 6). Nonetheless, the identified putative AMGs were carried by different viral species, which suggests that these AMGs encode essential metabolic functions that could be beneficial to prokaryotic hosts at both sites, and thus enhance viral fitness at both slope and bottom-axis CD locations.

Putative AMGs involved in assimilatory sulfate reduction were common in CD viruses with higher relative abundances in the bottom-axis samples, compared to the slope samples (Supplementary Fig. 5b and Supplementary Data 8). Among these AMGs, we identified nine putative *cysC* and *cysH* genes that participate in the reduction of sulfate to sulfite (Supplementary Data 10). AMGs coding for *cysH* were also recently reported in deep-sea viruses from the Southwestern Indian Ocean sediments[16].

Gene maps of representative CD viral contigs with co-occurrences of *cysC* and *cysH* and phage terminase genes within viral genomes are shown in Fig. 6a. To understand the origin of the putative CD AMGs related to sulfur assimilation, we recruited the top five (a) CysC proteins from the eggNOG database (v5.0) with close homology to our CD viral CysC proteins, and (b) CysC-encoding AMGs predicted from different viral data sets[7,15,46,48,58], respectively. The similarity between our CD CysC-encoding AMGs and those CysC proteins deposited in the eggNOG database (v5.0) ranged from 27% to 47% (Supplementary Data 10). These similarity percentages were lower when we compared our putative CysC-encoding AMGs with those identified in global-scale ocean viral data sets, including those from deep-sea sediments and permanently anoxic basins[7,15,46,48,58] (34% to 61%; Supplementary Fig. 7a). The phylogenetic analysis for three of our CD CysC proteins showed that they are distinct from their prokaryotic CysC homologs but cluster with CysC proteins from the different viral data sets referred to above (Fig. 6b). Similar phylogenetic results were

obtained for CysH proteins (Supplementary Fig. 7b and Supplementary Fig. 8).

The distinct phylogenetic results and the moderate similarity of the CD Cys proteins to those that are publicly available, prompted us to perform protein structure prediction for the CysC protein from the viral contig T1B8_NODE_1222 (Fig. 6c). We used the web-based Phyre2 tool that predicts protein structure and function using homology with known proteins available in protein data banks[60] (see Methods). The Phyre2 results predicted that CD CysC sequences belong to P-loop containing nucleoside triphosphate hydrolases and specifically to those hydrolases with a structural domain for adenosine-5'phosphosulfate kinase (APS kinase). The top three Phyre2 hits showed that 92–99% of the CysC protein sequences (109–135 residues) have been modeled with 99% confidence and exhibit structural homology with prokaryotic APS kinases. This suggests that CD CysC proteins could catalyze the phosphorylation of APS to 3'-phospho-APS, an intermediate step in sulfate assimilation. The putative AMGs related to assimilatory sulfate reduction could probably increase viral fitness in these CD sediments by enhancing the metabolic flexibility of the prokaryotic hosts as described elsewhere[58]. In addition, these AMGs might benefit hosts by ensuring an adequate supply of soluble thiolome pools (S-containing compounds such as amino acids and their intermediates) generated via sulfate assimilation, which can be utilized by hosts for heavy metal and metalloid detoxification (e.g., arsenic and mercury)[61,62]. Bio-accumulation of toxic metals has been detected in the water column of Mariana Trench[63,64], while our data indicate accumulation of heavy metals like mercury and arsenic in CD sediments[39], especially at the bottom-axis sites (Supplementary Fig. 9). AMGs related to sulfur metabolism are reported to be carried by viruses in various oceanic settings, including hadal realms[31,65], which could indicate that these AMGs can increase viral fitness under the different redox and nutrient conditions detected in CD. We also argue that the identified AMGs related to sulfur metabolism in Challenger Deep could enhance the heavy metal detoxification mechanisms of the prokaryotic hosts, and thus, increase viral fitness.

We identified 13 potential viral genes involved in the biosynthesis and accumulation of lipid A (e.g., *lpxA/D and*

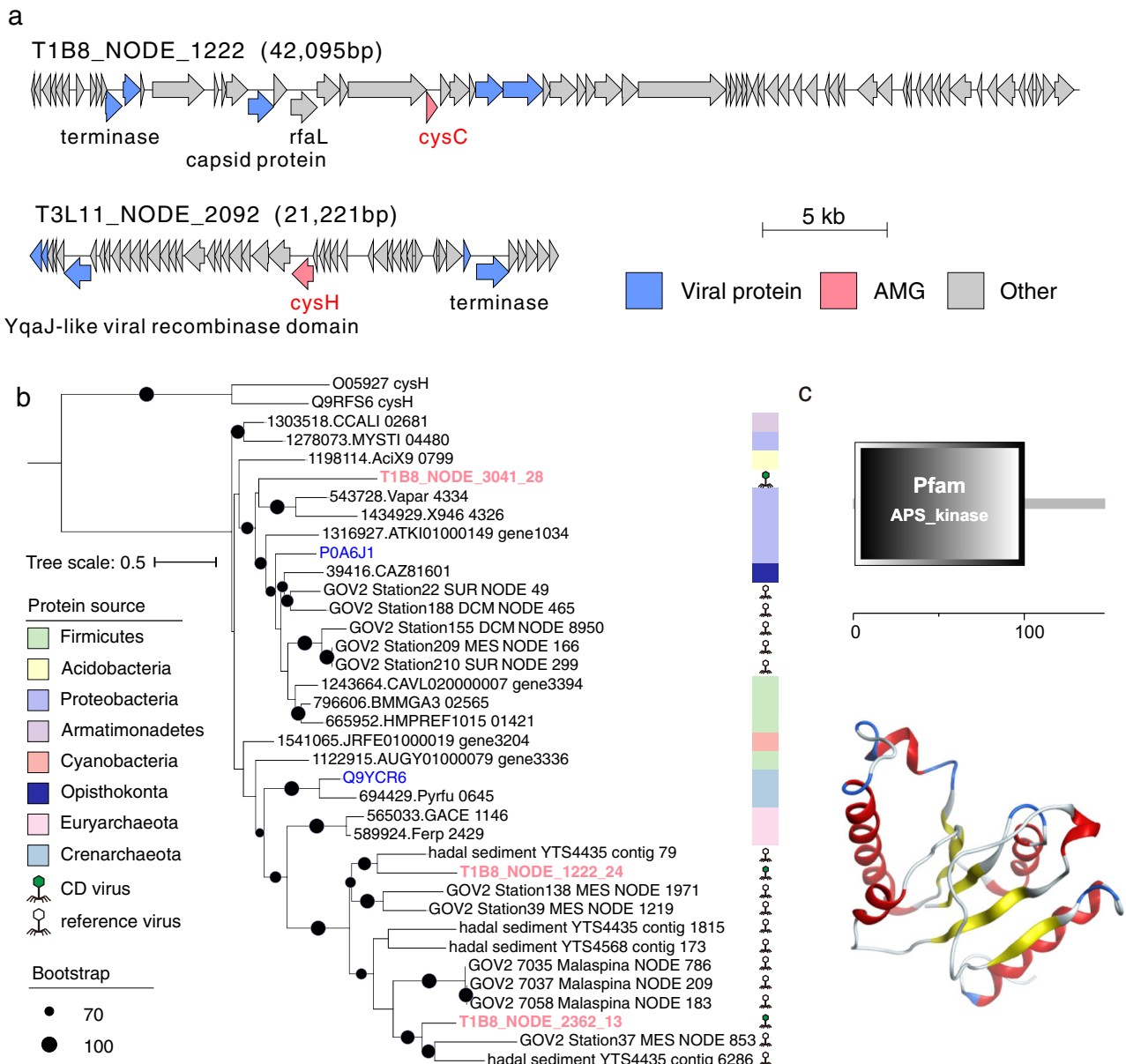

**Fig. 6 Genomic analysis of viral assimilatory sulfate reduction genes. a** Genome maps of two viral contigs containing sulfate reduction related genes. Genes encoding viral proteins were annotated by CheckV. AMGs, viral-specific and non-phage-like, and uncharacterized genes are shown in pink, blue, and gray, respectively. CysC/H and viral-specific proteins are labeled. Detailed function annotations of genes in viral contigs are listed in Supplementary Data 8. **b** Maximum-likelihood phylogenetic tree of CD CysC proteins. The CysC proteins predicted in CD viral genomes were used to construct a phylogenetic tree using homologous CysC proteins deposited in the eggNOG database (V5.0) and publicly available viral data sets. CysH proteins were used as an outgroup. We also included two CysC homologs from the Uniport database (in blue) with experimental evidence of function at the protein level. Bootstrap values (1000 replicates) ≥70% are indicated at nodes. **c** Functional domain and three-dimensional structure of CysC protein (T1B8_NODE_1222_24).

*kdsB*)[66,67], and maturation of lipopolysaccharides (LPS) (e.g., *WaaE/F/L*)[68–70] (Supplementary Fig. 10 and Supplementary Data 8). T4-like phages were reported to encode LPS biosynthesis genes, which might alter the surface composition of the infected host to prevent multiple phage infections, or may simply act as 'stuffer DNA' for headful packaging in phages with a large genome like cyanophages[71,72]. The potential viral LPS genes identified in CD viruses are not cyanophage-related genes, and thus their role is unclear. We suggest that they could be involved in cell membrane stability[73], and potentially benefit viral fitness by enhancing the structural and mechanical role of the host's outer membrane that is exposed to extreme hydrostatic pressures (>1000 atm) at these hadal depths. Putative viral metabolic genes

related to membrane biogenesis (e.g., cytidylyltransferase) were detected in viruses from hydrothermal sediments and it was suggested that they might enhance viral fitness by regulating the host's membrane fluidity and phospholipid homeostasis[74].

We also identified 5 putative viral metabolic genes involved in rhamnose biosynthesis in the CD viruses (Supplementary Data 9). Large DNA viruses like chloroviruses and prasinoviruses are known to synthesize rhamnose[75,76]. However, chloroviruses and prasinoviruses are primarily known to infect algae[77,78], and were absent from our CD viruses. The putative *rmlB/C* AMG recovered from the CD sediments was found in complete circular viral contigs and was flanked by viral-specific genes (Supplementary Fig. 10). The protein similarities of the viral *rmlB/C* sequences

with those from closely related proteins available in public databases ranged from 27% to 51% (Supplementary Data 10). Rhamnose-containing cell wall polysaccharides are considered phage receptors[79,80], while rhamnose operons are reported to affect bacterial motility and biofilm formation[81].

Whether the putative genes related to LPS and rhamnose biosynthesis could enhance host cell membrane flexibility or regulate the host's ability for biofilm formation in hadal surficial sediments requires further investigation. Overall, we suggest that the possible benefit of the putative AMGs to CD viruses might depend on whether the viruses are lytic or lysogenic[82], and the lytic state of lysogenic viruses is influenced by environmental conditions[83]. Finally, viral elements can also appear as prophages that upon infection can replicate and produce viral particles without destroying the host cell[84].

## Conclusions

Metagenomes generated from hadal sediments (>6 km depth) collected along the V-shaped Challenger Deep in the Mariana Trench, describe previously unidentified hadal viral communities that display a discrete separation along the trench. This discrete separation appears to be influenced by available nutrient sources that shape the prokaryotic (host) community structure between slope and bottom-axis sites. The presence of potentially lytic viral communities in CD may enable the high microbial density detected in this otherwise nutrient-poor hadal realm via the viral shunt that affects the in situ availability of labile carbon in those sediments. Future work will benefit from high-throughput culturing experiments of hadal viruses, as well as host-virus interaction experiments that can reveal the metabolic potential, viral shunt efficiency, and virus-host interaction networks in hadal nutrient cycling. The factors that control the biogeochemistry of the Challenger Deep sediments, including anthropogenic impacts, and that shape the metabolic and functional adaptions of hadal viruses and microbes will be topics of future research.

## Methods

**Sampling**. Sediment cores from 13 CD sites were collected at water depths between 5400 m to 10911 m with three hadal cruises (Dayang37, Tansuo01, and Tansuo03) at the Challenger Deep in 2016–2017. The sediment cores were immediately sectioned into 2 or 3 cm layers, and stored at −80 °C for nucleic acids (DNA and RNA) extraction, and viral metagenomic analysis (Supplementary Table 1).

**Nucleic acids extraction, metatranscriptome, and metagenome library preparations**. We selected 37 sediment layers from different sediment cores covering slope and bottom-axis samples (Supplementary Table 1), and extracted DNA from 10 g~40 g using the PowerMax soil DNA isolation kit (MoBio, Carlsbad, CA, USA) following the manufacturer's instructions. DNA concentrations were measured using a Qubit™ 2.0 Fluorometer (Invitrogen, Carlsbad, CA, USA). Samples with <2 ng μl$^{-1}$ of DNA were concentrated using AMPure XP beads (Beckman Coulter, CA, USA) before the preparation of the libraries. The extracted DNA was sheared randomly using ultrasonication (Covaris M220, 200 cycles per burst for 65 s or 45 s) and was used to prepare DNA libraries with insertion sizes of ≥350 bp and up to 550 bp using the TruSeq Nano DNA Sample Prep Kit (Illumina, San Diego, CA, USA). For negative controls, we included two blanks that were treated and processed the same as the sediment samples. We concentrated each control DNA with AMPure XP beads (Beckman Coulter, CA, USA). The DNA concentrations of our control samples yielded <2 ng in a total volume of 10 ul, which was far less than the DNA input (100 ng) recommended by TruSeq Nano DNA Sample Prep Kit (Illumina, San Diego, CA, USA). Due to this constraint we prepared the libraries of our two controls using the TD503 kit (Vazyme, Nanjing, China) with an insertion size of 350 bp.

Total RNA was extracted from 10 g of three sediment layers (6–9, 12–15, and 18–21 cmbsf) collected from the T3L11 site (10,908 m depth) using a PowerSoil Total RNA Isolation Kit (MoBio, Carlsbad, CA, USA) following the manufacturer's instructions. The RNA extracts were treated with TURBO DNase (Invitrogen, Waltham, MA, USA) to remove genomic DNA. The absence of carryover DNA was confirmed with PCR reactions using prokaryotic primers for the V3–V4 region 341 F (5'-CCTAYGGGRBGCASCAG-3') and 802 R (5'-TACNVGGGTATCTAATCC-3'). Each 50 μl reaction contained 1.25 U PrimeSTAR HS DNA Polymerase (Takara, Japan), 5× PrimeSTAR Buffer (Takara, Japan), 200 mM dNTPs (Takara, Japan,

dNTP Mixture), and 0.3 μM of each primer (final concentrations). The PCR reactions were performed at 94 °C for 10 s, followed by 35 cycles of 98 °C (10 s), 55 °C (10 s), and 72 °C (30 s). We used the Ovation® RNA-Seq System V2 Kit (NuGEN, San Carlos, CA, USA) to make double-stranded cDNA (ds-cDNA) from 1 ng total RNA with random primers. The ds-cDNA was used to prepare the metatranscriptome libraries as described in the metagenome library preparation.

**Metagenome sequencing and assembly**. Libraries were sequenced using 300 bp paired-end reads at Miseq platform or 150 bp paired-end reads at Novaseq 6000 or Hiseq 2000 platform. Fastp (v.0.20.0)[85] was used to remove adapter and low-quality reads (assigned by >20% of the read length have quality score <20 or read length <50) with parameters (-w 16 -q 20 -u 20 -g -c -W 5 −3 -l 50). For metatranscriptomes, reads that mapped onto the rRNA sequences by SortMeRNA (v.2.1)[86] and sequences in an in-house contaminant database (including sequences of mouse, human and common laboratory contaminant bacteria genomes[87] downloaded from NCBI) by Bowtie2(v.2.4.1)[88] with setting -N 1 were discarded. The high-quality metagenome reads for each site (MC02, D1T1, D1T2,T1B3, T1B5, T1B8, T1L6, T1B10, T1B11, T1L10, T3L8, T3L11, and T3L14) were merged for assembly using SPAdes (v3.13)[89] with a $k$-mer set of 21, 33, 55, 77, 99 and 127 under the '--careful' mode to achieve the best assembly results for low-abundance microbial groups[90]. Contigs ≤10,000 bp were removed prior to viral identification.

**Identification, decontamination, and classification of CD viruses**. We identified viral sequences from metagenomic assemblies in four steps using the published standards[91], and the following enhancements: 1. Metagenomic contigs (>10 kb) from co-assembled metagenomes of each site were processed with the viral identification tools wrapped in What the Phage (Version 1.0.1, setting: --filter 10000 –identify[40]; tools: MARVEL[92], VirFinder[93], PPR-Meta[94], VirSorter[95], MetaPhinder[96], DeepVirFinder[97], VIBRANT[50], VirNet[98], Phigaro[99], Virsorter2[100], and Seeker[101]) to obtain putative viral contigs. 2. We annotated the predicted putative virus contigs using the eggNOG database (v5.0.0)[102]. Previous studies retained those contigs as putative viral if they contained hallmark viral genes including those contigs with viral sequences that had high percentages (e.g., ≥80%) of genes of unknown and hypothetical function[103,104]. Similarly, we retained those contigs as putative viral if they contained ≥2 virus-specific genes (annotation contains words from the list: "capsid", "phage", "terminase", "base plate", "baseplate", "prohead", "virion", "virus", "viral", "tape measure", "tapemeasure neck", "tail", "head", "bacteriophage", "prophage", "portal", "DNA packaging", "T4", "p22", and "holin")[105], or contained viral sequences and had ≥70% of proteins assigned as hypothetical protein, unknown function or Viruses. 3. The retained contigs that contained prokaryote-specific genes (e.g., ribosomal genes) were further removed. 4. CheckV (v.0.8.1)[44] was used to assess the quality of all putative viral contigs and to detect the viral-host boundaries for subsequent removal of host region from provirus. Contigs without determined completeness and viral-specific genes (as predicted by CheckV[44]) must contain viral signatures using benchmarked viral prediction tools (DeepVirFinder, VirSorter, VirSorter2, MARVEL, and VIBRANT) with conservative cutoff, published in standard operating procedures[91]. The putative viral contigs that remained after applying the four steps explained above were considered high-confidence viral contigs in this study.

Bowtie2[88] was used to map reads from the two blank controls. Viral contigs mapped with ≥1 read(s) from the controls were considered potential contaminants. This resulted in the removal of ten viral contigs from further analysis.

All positive and host-contamination-free viral contigs (simplified as bona fide viral contigs) were clustered into vOTUs with cd-hit-est (v4.8.1, setting: -c 0.95 -aS 0.85 -n 10 -d 0)[106] at species level, based on >85% alignment of the smallest contigs at 95% average nucleotide identity[41]. We adopted a previously described majority-rules approach to assign viral family[7]. In brief, we used blastp (version 2.9.0+) to query all proteins from CD vOTUs against NCBI viral RefSeq database release 208 (https://ftp.ncbi.nlm.nih.gov/refseq/release/viral/, downloaded on 4 January 2022). A vOTU was assigned as a family if ≥50% of its proteins hit to family level with a bitscore ≥50. The remaining unassigned vOTUs were classified using Demovir pipeline (https://github.com/feargalr/Demovir)[107] with default settings. Demovir pipeline searched proteins from CD vOTUs against a redundant viral subset of the TrEMBL database (cluster at 95% identity; https://www.uniprot.org/downloads). We assigned families to unassigned viral contigs only if the similarity of the proteins to a homolog for one taxon reached ≥50%.

**Comparison to viruses from other data sets**. We compared the 1628 identified CD vOTUs at species and genus level using vConTACT2[45] with viral contigs from published databases, including: (i) hadal and non-hadal deep-sea sediments[15,30] ($n = 7305$); (ii) wetland sediments[46] ($n = 1212$); (iii) thawed permafrost[47] ($n = 1896$); (iv) seawater (Global Ocean Virome 2.0, $n = 195,728$)[7] and (v) reference sequences (Prokaryotic Viral RefSeq85-ICTV, $n = 1825$). For species level, cd-hit-est (v4.8.1, setting: -c 0.95 -aS 0.85 -n 10 -d 0) was used to identify CD vOTUs as novel (nucleic acid similarity <95%). Prodigal v2.6.3[108] was used to predict open reading frames (ORFs) for all viral contigs used for comparisons, and the predicted protein sequences were imported to vConTACT2[45] for identification of previously undescribed viral clusters (genus level).

**CD gene annotation and metabolic genes analysis**. CD viral genes in viral contigs were predicted by Prodigal[108]. All predicted proteins were annotated against eggNOG v5.0.0 database by eggnog-mapper with the default setting. High-confidence viral contigs were imported to DRAM-v (v.1.2.4)[109] and VIBRANT (v.1.2.1)[50] for annotation and identification of AMGs based on KEGG, Pfam, UniRef90, dbCAN, RefSeq viral, VOGDB (including pVOGs) and MEROPS databases, using default parameters. We removed AMGs identified from VIBRANT[50] with T, B flag, auxiliary scores >3. We also removed those AMGs that were not included in the manually curated list of potential AMGs that we compiled after the DRAM-v[109] results (see below). All remaining putative AMGs were further checked for their position on contigs to ensure their viral origin. We retained only those manually curated AMGs that contained clearly-identified viral genes (Supplementary Data 9).

Further analyses were also conducted on the key AMGs identified by VIBRANT[50]. Viral AMGs assigned as K00860 (cysC, APS kinase gene) were identified in the viral genomes, and they were compared to protein sequences from eggNOG v5.0.0 database and publicly available viruses (blastp, $10^{-5}$ for E value) to recruit relevant reference sequences. We recruited the top 5 reference CysC sequences in eggNOG v5.0.0 database and publicly available viruses, respectively, for each identified viral CysC protein. We aligned all CysC protein sequences (CD viral CysC proteins, CysC reference proteins from eggNOG database, and two CysC sequences with experimental evidence at protein level from the Uniprot database) and two CysH protein sequences as outgroup with Mafft (v7.453, setting: --maxiterate 1000 -localpair)[110], and filtered them with TrimAL (v1.4.rev15[111]; default parameters) to remove poorly aligned columns. Maximum-likelihood phylogenetic trees were reconstructed using iqtree (v2.0.3, setting: -m MFP -bb 10000), which automatically searched the best-fit partition model before tree reconstruction[112]. The resulting newick file with bootstrap value was uploaded to iTOL v4[113] for visualization. The structure prediction for CysC protein was performed with the web-based Phyre2 tool[60]. Structural homologies were analyzed using models generated by Phyre2 using a confidence threshold of >98%, and identity threshold of >29%. The accuracy of the models constructed using Phyre2 is described as extremely high when the sequence identity is above 30–40%. However, lower sequence identities can be equally accurate and useful as long as the confidence threshold is high, which was the case in our examined CysC proteins. The functional domain for CysC was identified and annotated by SMART[114]. This workflow was also applied to analyze other key viral metabolic genes.

**Prediction of virus and host linkages**. We collected 586 MAGs[39] recovered from the metagenomes used for prediction of viruses in this study. Virus-host interaction was predicted by four different in silico methods[47,51] that include: 1. search for sequence homology by calculating the nucleotide identity of vOTUs and prokaryotic MAG sequences using BLASTn (v2.9.0+, setting: -evalue 0.001 -perc_identity 70). The conditions for retaining positive matches were (a) alignment length ≥2500 bp, (b) minimum nucleotide identity ≥70%, and (c) alignment coverage <90% of host sequences[5]. 2. search for matches between viral contigs and host CRISPR. Spacers and repeats of host were predicted in clean reads of metagenomes using crass (v1.0.1)[115] using default settings, and were extracted using crisprtools (https://github.com/ctSkennerton/crisprtools). Spacers and repeats in assemblies of metagenomes were predicted by MinCED (v.0.4.2)[116]. The combined spacers from reads and assemblies were compared to sequences of viral contigs using BLASTn (v2.9.0 + ). We retained matches that contained ≤1 mismatch and had an E value of ≤$10^{-5}$. For each spacer with a match in any genome of CD vOTU, the repeat sequence from the same assembled CRISPR region was compared to all prokaryotic population genomes via BLASTn (v2.9.0+, 100% nucleotide identity and E value of >$10^{-10}$). We performed this step to link the assembled CRISPR region (and, therefore, any viruses matching spacers in that CRISPR region), to a host. 3. search for viral and host tRNA genes using tRNA-scan (v. 2.0.7)[117]. Only exact matches were considered. 4. search for similarity of $k$-mer frequencies using WIsH[118] with default parameters to predict the potential host of a query virus. Sequence similarity-based matches were also manually curated to avoid false positive results caused by viral contigs binned into prokaryotic genomes.

All the prediction results generated using the four steps above were scored by VirMatcher. VirMatcher (https://bitbucket.org/MAVERICLab/virmatcher/src/master/) is the only available tool that provides a confidence score for host predictions. High-confidence predictions (scores ≥3) were used to assign hosts in this study. The coverage of hosts (species level) in each metagenome was calculated by CoverM v0.6.1 (parameters: -m trimmed_mean --min-read-percent-identity 0.95 --min-read-aligned-length 50 --proper-pairs-only --min-covered-fraction 0.1). The relative abundance of a host was calculated by the coverage of the host divided by the total coverage of all MAGs (species level).

**Viral abundance and diversity**. To calculate the coverage (sequencing depth) of each viral contig, clean and qualified reads from each sample were mapped against all CD viral contigs using BWA (v 0.7.17)[119] and sorted with samtools (v1.9)[120] to generate a bam file. CoverM v0.6.1 (https://github.com/wwood/CoverM) was used to filter low-quality mappings in bam file and generate the abundance profiles for samples (parameters: contig mode for each viral contig, -m mean --min-read-percent-identity 0.95 --min-read-aligned-length 50 --min-covered-fraction 10). Contigs with sequencing coverage rate <10% were reported as having zero

sequencing depth. Normalized coverage values (sequencing depths per giga base sequencing data) were used to represent relative abundances of viruses for comparison across metagenomes. The normalized coverage was multiplied by 10 and ceiled to represent the times of one species present in each sample for the calculation of the ACE and Chao1 indices. The metatranscriptome reads were also mapped to viral contigs/genes using BWA (v 0.7.17)[119] and were then counted with aligned length ≥50 bp and identity ≥95% by CoverM v0.6.1 (parameters: --min-read-percent-identity 0.95 --min-read-aligned-length 50) to generate abundance profile.

**Statistics and reproducibility**. All statistical analyses were performed using R (v.4.0.4). We used *vegan* (v.2.5–7) in R to calculate the alpha and beta diversities of viral communities and Bray–Curtis distances for vOTUs abundance profiles. Pheatmap package (v.1.0.12) was used to cluster metagenomes in heatmap. For comparing the vOTUs between slope and bottom-axis sites, we used Shannon, Simpson, ACE, and Chao1 indices. The detected vOTUs were tested using the Wilcoxon test with the *ggpubr* package (v.0.4.0).

**Reporting summary**. Further information on research design is available in the Nature Research Reporting Summary linked to this article.

## Data availability

The raw reads of CD metagenomes and metatranscriptomes and MAGs were deposited in GenBank under BioProject number PRJNA635214. The DNA sequences of the 1628 viral contigs were deposited in NCBI GenBank under accession number JAODGZ000000000. The marine viruses-related data sets utilized in this study are cited in Supplementary Table 2. Source data for figures also provided in Supplementary Data 11.

## Code availability

All software and R packages used are open source and described in the Methods section. No custom code was used to analyze data in this study, and further details are available on request.

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

## Acknowledgements

We give special thanks to the members of the R/V DY37, TS01, and TS03 for their invaluable efforts in the sampling cruises. We thank J. Li, S.X. Wang, Y.Z. Xin, J. Chen, and D.S. Cai for their skillful handling of the lander and sediment sampler. We also thank the supercomputer center of Sanya University. This research was supported by the Hainan Provincial Natural Science Foundation of China (No. 322CXTD531).

## Author contributions

Y.Z. and Y.W. conceived and designed the study. Y.Z. conducted bioinformatic analyses and results visualization. Y.Z. conceived and generated Fig. 1. Y.Z. and P.M. analyzed data and summarized the results. Y.Z. and P.M. drafted the manuscript. V.P.E., D.V., M.B.S., and Y.W. critically revised the manuscript.

## Competing interests

The authors declare no competing interests.
