## [Peer Review File · Communications Biology]

Reviewers' comments:

Reviewer #1 (Remarks to the Author):

This study analyzed 13 sediment cores from the Challenger Deep and identified 1,628 vOTUs. Network based analyses indicated the novelty of these vOTUs compared to several reported ecosystems. Higher viral diversity of bottom samples than slope samples were observed. The results of virus-host link and viral encoded AMGs suggested the potential roles of viruses in mediating the carbon, nitrogen, and sulfur cycles, and improving the adaption abilities of their hosts. This study was well organized, and data was analyzed using proper methods. My major concern is that the sediment viruses of Challenger Deep had been reported by Zhao et al., (2022) recently. the authors should compare with their data and highlight the new founding in the present study.

Specific comments:

As you use the viral contigs after manual curation (1628 vOTUs), I'd like to know the performance comparison of the tools about the high confidence viral contigs. Was there any software identifying all high confidence vOTUs?

Line 37 slope-axis should be mentioned in abstract, otherwise we do not know what the bottom-axis was compared with.

Line 83-113 Please briefly introduce what you did in this paragraph, not an extended edition of abstract.

Line 120 I'd like to know the relative transcription level of viruses compared to hosts (something like certain prokaryotic housekeeping gene?), even though the percentage might be very low.

Line 137 I cannot understand this sentence.

Line 142 The gene number of each virus is different and two genes for all viral contigs is not appropriate. Advised to use percentage.

Line 227 Please include the relative abundance of the potential host MAGs. The discussion about the influence of lytic viruses in hadal carbon and nitrogen cycling are based on the widespread and high abundance of these potential hosts.

Line 277 This conclusion is too strange considering you only predicted the hosts of 14 vOTUs.

Line 456 Annotated using which database? eggnog?

Line 471 Please include the standard considering a viral contig as contamination. The coverage or the covered percentage?

Line 477 Considering you performed vConTACT2, please include the taxonomy assignment results of this software.

Line 553 "a viral contig database"?

Line 558 The threshold value of 10% is too low, which might cause too many false data. Please consider the threshold value of vOTUs clustering (85%).

Line 544 check grammar.

Many commas were missing in the manuscript, such as Line 129, 192, 222, 410.

Reviewer #2 (Remarks to the Author):

The authors took on the endeavor to detect, characterize and compare viruses recovered from deep ocean trenches. I appreciated all the efforts to study these under-sampled environments that will advance our understanding of environmental viruses and their ecological functions. Overall, the topic is important, and the results are exciting to read. However, I do have some major concerns.

1) More thoughts are needed in methodology: example 1: the authors applied one unpublished or not peer-reviewed workflow to detect viral contigs. Although the tools mentioned in the workflow are widely used, the cut-offs and ways of sorting the results are encrypted in the workflow; example 2: suspicious methods to screen putative viral contigs such as 'at least 70% of proteins in the contig were assigned as 'hypothetical protein', 'unknown function'; example 3: identify lytic viruses using

VIBRANT that can lead to misinterpretation of the results.

2) The authors need to be careful when citing references to support your discussion. Examples: citing soil viruses for supporting low lysogeny; citing thawed permafrost papers for permafrost ecosystem; citing Paez-Espino et al. (2017) for the suspicious method of screening putative viral contigs.

More detailed comments are attached.

Reviewer #3 (Remarks to the Author):

The manuscript "Ecogenomics reveals novel viromes in the deepest ocean trench on Earth" describes a metagenomic survey of viral populations in the hadal region of the Marianas trench. The authors describe the analysis of metagenomes to identify viral OTUs, analyze these for species distribution, predicted hosts, and potential auxiliary metabolic genes. Though the study presents interesting data about the viral populations of the deepest regions of ocean floor it is very descriptive, with not a lot of evidence supporting the conclusions beyond the observations described. The paper could be improved by addressing the following comments.

1. The term 'ecogenomics' is used in the title but never defined or discussed in the text of the paper.

2. The first use of the term 'vOTU' (line 107) isn't preceded by a definition of what that means.

3. In the Results section (lines 131-136) the curation process yields 1628 contigs, but the authors then state that these are further separated into 1628 vOTUs >10kb and 6 < 10kb (i.e. more than the number of contigs). This should be clarified.

4. Figure 2a is very confusing and needs (at least) to be better described and have better labels. It's not clear what the X axis represents (length of contig maybe?). It seems that the dots and lines under the histogram might represent overlap of different methods – though it's not clear how that's being represented or what it means.

5. Reference to Figure 1 in line 149 is a bit confusing – I guess it's there because the different habitats have just been described?

6. Better labeling of the X axis on Figure 3 would be helpful – it's very hard to interpret as is without reading each of the sample labels. Bars or other graphic indicating groups of samples (trough vs. slope, e.g.) would be useful.

7. Lines 200-202: it's not clear how the distribution of viral populations is consistent with distributions of prokaryotic communities. Needs clarification.

8. Figure 4b should have a visual key in the figure to help readers remember the meaning of blue and red bars.

9. Line 220-221. "These prokaryotic MAGs were recovered largely... from the same metagenomes as the viral contigs." Is confusing – it implies that there are MAGs that were from sources other than the same metagenomes (which I assume there were not)

10. Lines 277-278: "most CD viral populations target" – this is way overstating things given that the number of vOTUs that had predicted hosts was very low. Same with next sentence too-

11. Lines 320- . The structural modeling comes out of nowhere - it makes sense, it's just not adequately described in the results or methods. Nor is the conclusion that the CD CysC can carry out the function (is this the normal function of CysC? What more did predicted structure show?)

12. Lines 332: how is the statement "our data indicate accumulation of heavy metals" supported? Was this from other measurements taken of the sediments? If so this needs to be described more fully here.

13. Line 338-339: "increase the viral fitness towards the potential toxic effect of the arsenic accumulation." Doesn't make sense.

14. A comparison of AMGs from another set of metagenomes would be helpful: are these interesting observations or just what's seen everywhere?

We thank the 3 reviewers for their helpful comments and suggestions on the manuscript which we find is now greatly improved thanks to this input. Please find below our point-by-point responses. Line numbers in the responses refer to line numbers on the tracked changes version when using the setting of Simple Markup

Reviewer #1 (Remarks to the Author):

Q1. This study analyzed 13 sediment cores from the Challenger Deep and identified 1,628 vOTUs. Network based analyses indicated the novelty of these vOTUs compared to several reported ecosystems. Higher viral diversity of bottom samples than slope samples were observed. The results of virus-host link and viral encoded AMGs suggested the potential roles of viruses in mediating the carbon, nitrogen, and sulfur cycles, and improving the adaption abilities of their hosts. This study was well organized, and data was analyzed using proper methods. My major concern is that the sediment viruses of Challenger Deep had been reported by Zhao et al., (2022) recently. the authors should compare with their data and highlight the new founding in the present study.

R1: We thank this reviewer for the overall positive comments, and we appreciate his/her concern. However, we need to that point out that the viral communities reported by Zhao et al., (2022) are from the upper slope of the trench, and specifically, from surficial sediments (0-18 cm) at four sampling sites between 5.4 to 6.7 km water depth (see Table S1 by Zhao et al., 2022). In contrast, our study reports data on viral communities from 13 sampling sites that cover both slopes and the bottom-axis of the Challenger Deep, targeting primarily hadal depths below 6 km, with the deepest sampling depth at ~11 km.

Although Challenger Deep is still understudied, we know from literature that the funneling effect and geographic isolation creates heterogeneity between sites along the slopes of the trench and its bottom axis, shaping prokaryotic microbial communities and diversity (e.g., Luo et al., 2017 Zhou et al., 2022). Because of this we believe that even intra-trench comparisons of microbial and viral communities (e.g., slope vs. slope sites or bottom-axis vs. bottom-axis sites) will produce new findings. The inclusion of a greater number of sites as well as sites that extend down to ~11km in our study therefore does not just extend the findings of viruses in the Zhao et al. 2022 study. Nonetheless, to address this reviewer's concern, we compared our viral data to the data reported by Zhao et al. 2022, and found that 98% are new species (<95% identity in 85% of sequence length) and 76% are new genera (estimated by vcontact2). We amended one sentence in our abstract to say "Here, we collected 13 sediment cores from slope and bottom axis sites across the Challenger Deep (down to ~11 km depth), and identified 1,628 previously undescribed viral operational taxonomic units at the species level."

And in the main text in lines 159-163 we now say: "Our CD vOTUs were also novel when compared with viromes identified at the hadal slope sediments of the Mariana Trench⁴⁸. Specifically, 98% of our CD viral contigs were new species (< 95% identity in 85% of sequence length) and 76% of them were new genera (estimated by vcontact2), when compared with the identified viruses from the upper slope (5.4-6.7 km depth) of the trench⁴⁸."

For reviewer's convenience the papers referred to above are here:

Luo M, Gieskes J, Chen L, et al. Provenances, distribution, and accumulation of organic matter in the southern Mariana Trench rim and slope: Implication for carbon cycle and burial in hadal

trenches[J]. *Marine Geology*, 2017, 386: 98-106.

Zhou Y, Mara P, Cui G, et al. Microbiomes in the Challenger Deep slope and bottom-axis sediments[J]. *Nature Communications*, 2022, 13(1): 1-13.

Zhao J, Jing H, Wang Z, et al. Novel Viral Communities Potentially Assisting in Carbon, Nitrogen, and Sulfur Metabolism in the Upper Slope Sediments of Mariana Trench. *mSystems*. 2022;7(1):e0135821.

Specific comments:

Q2: As you use the viral contigs after manual curation (1628 vOTUs), I'd like to know the performance comparison of the tools about the high confidence viral contigs. Was there any software identifying all high confidence vOTUs?

R2: We performed laborious manual curation of the viral contigs to avoid false-positive predictions, and PPR-meta analysis to identify the most confidently-assigned viral contigs by the twelve tools incorporated into our pipeline (Figure 2a). We now provide Supplementary Table 4 that summarizes the screening criteria used for identifying the viral contigs, and the cut-offs used for each tool. We didn't a single individual software to identify the high confidence vOTUs. We now include the term "putative" viral contigs in various parts of the paper after reviewer's 2 request.

Q3: Line 37 slope-axis should be mentioned in abstract, otherwise we do not know what the bottom-axis was compared with.

R3: This reviewer is right. We rephrased accordingly and now reads "Community-wide analyses revealed distinct viral diversity across the trench which is significantly higher at the bottom-axis, when compared to slope sites. In silico predictions indicate viral infection of key prokaryotes involved in hadal carbon and nitrogen cycling". Please see lines 36-38.

Q4: Line 83-113 Please briefly introduce what you did in this paragraph, not an extended edition of abstract.

R4: Lines 82-113 describe environmental features of Challenger Deep (e.g., geochemistry, in situ temperature/pressure conditions, carbon distribution) and summarize what is known for hadal microbial communities. This information is not described/discussed in the abstract and we consider it of importance to the reader. We also believe that lines 82-84 and 105-113 that describe the sampling sites and the aims of this study do not overlap with what is already discussed in abstract.

Q5: Line 120 I'd like to know the relative transcription level of viruses compared to hosts (something like certain prokaryotic housekeeping gene?), even though the percentage might be very low.

R5: This is a really interesting comment. Our data allowed us to predict hosts for a small fraction of the CD viral community (14/1628 vOTUs; see lines 230-232), and we also have only three metatranscriptome libraries (6-9, 12-15, 18-21 cmbsf) from one sediment core collected at ~11 km depth. Most genes were not mapped to by metatranscriptome reads in our libraries. For those transcripts that mapped, a significant fraction was characterized as genes of unknown function (16% of the total mRNA pools; please see Supplementary Data 4, in Zhou et al., 2022). We do not consider this surprising since the available mRNA data from hadal trenches are limited, and we also know that RNA can be readily degraded especially when extensive recovery times of samples occur (~5

hrs from 11 km depth to surface; this study). This would make it challenging even for genes from the core genome (e.g., housekeeping genes) to be reliably used for any inter-sample comparison. To address this comment, we listed the number of host genes that were mapped by metatranscriptome reads in the table below:

Host	Number of host genes	Number of host genes mapped by metatranscriptomes		
		T3L11(6-9cmbsf)	T3L11(12-15cmbsf)	T3L11(18-21cmbsf)
B51T1B5	1820	2	1	3
B10T1B11	1693	0	0	2
B44T3L14	1475	27	9	84
B17T3L8	2057	1251	1037	1617
B52T3L11	1919	52	44	176
B12T1B11	2275	82	65	209
B10D1T1	1335	37	30	116
B46T1B5	1827	1	1	3
B31T1L10	4066	534	383	1710
B94T1B8	2217	3	2	13
B58T1B8	2534	2	2	4
B15D1T1	2780	5	2	11
B22T3L8	2986	1	0	12
B3D1T2	2967	9	4	23
B9T3L8	2734	114	73	529
B1T1B11	2458	327	214	1248
B60T3L14	2587	490	352	1502
B56T1B8	2732	138	105	653
B21D1T1	2635	47	32	217
B73T1B5	2032	69	48	332
B13T1B5	2185	680	521	1291
B93T1B8	2319	29	25	89
B64T3L11	2325	1765	1545	2159
B42T1B8	2325	151	111	436
B2T1B3	2368	12	11	40
B29T3L14	3069	631	477	1640
B16D1T2	2771	78	64	131

Q6: Line 137 I cannot understand this sentence.

R6: We apologize for that. Now on lines 136-138 it reads: “The degree of completeness and contamination of the CD vOTUs was estimated by comparing the sequences using CheckV against a large database of environmentally diverse complete viral genomes”.

Q7: Line 142 The gene number of each virus is different and two genes for all viral contigs is not appropriate. Advised to use percentage.

R7: We agree with the reviewer on this. However, providing percentages might not be as precise or widely accepted for estimating viral activity especially when bulk metatranscriptomic pools are used.

Nonetheless, to follow this reviewer’s advice, we now provide Supplementary Table 6 that shows the different % of genes mapped by each vOTU using metatranscriptome reads, and the different percentages of potentially active genes in the viral contigs (see below). We also rephrased the text (see lines 143-145) to clarify that we use 20% as threshold for identifying potential active viral contigs.

Supplementary Table 6: Different percentages of genes mapped in each virus using metatranscriptome reads to determine potentially active viruses

% of genes mapped in each virus using metatranscriptome reads	5%	10%	20%	30%	40%	50%	60%	70%	80%	90%	100%
Number of potentially active contigs	592	477	357	271	224	180	133	96	65	38	11
potentially active contigs (%)	36%	29%	22%	17%	14%	11%	8%	6%	4%	2%	1%

Q8: Line 227 Please include the relative abundance of the potential host MAGs. The discussion about the influence of lytic viruses in hadal carbon and nitrogen cycling are based on the widespread and high abundance of these potential hosts.

R8: We added a sheet to Supplementary Table 7 to show the relative abundance of the potential host MAGs in each metagenome (please see sheet “Relative abundance”), and we also provide this information in the text. Now on line 235-239 it reads: “These taxa include heterotrophs (e.g., Proteobacteria) and chemoautotrophs (e.g., Thaumarchaeota, Planctomycetota) involved in nitrogen and carbon cycling whose taxonomic signatures were abundant in CD sediments ³⁹, but with different relative abundances (7% to 43%) across the discrete sampling sites (bottom-axis vs. slope) (Supplementary Table 9).” See lines 572-576 for the method for estimate of the relative abundance of host MAGs.

Q9: Line 277 This conclusion is too strange considering you only predicted the hosts of 14 vOTUs.

R9: We agree with the reviewer and we rephrased accordingly. Now on line 280-283 it reads: “The predicted potential prokaryotic hosts for the 14 vOTUs may suggest that CD viruses target specific prokaryotic hosts in these CD sediments, however, this requires further investigation considering that our host predictions were accomplished for ~1% of the viral population that we identified.

Q10: Line 456 Annotated using which database? eggnog?

R10: Yes, we used the eggNOG database for annotation. We added text to clarify this and it now reads: “2. We annotated the putative virus contigs using the eggNOG database.” See lines 463-464.

Q11: Line 471 Please include the standard considering a viral contig as contamination. The coverage or the covered percentage?

R11: We added text to clarify. Now it reads (lines 482-484): “Viral contigs mapped with ≥ 1 reads from a blank control were considered potential contaminants. This resulted in the removal of ten viral contigs that were excluded from further analysis.”

Q12: Line 477 Considering you performed vConTACT2, please include the taxonomy assignment

results of this software.

R12: vConTACT2 clustered/classified only one viral contig using its reference database. This did not change the results of the taxonomy annotation that we already had.

Q13: Line 553 “a viral contig database”?

R13: We apologize for this vague statement. We were referring to all CD viral contigs. We rephrased lines 578-580 as follows: “To calculate the coverage (sequencing depth) of each viral contig, clean and qualified reads from each sample were mapped against all CD viral contigs using BWA (v 0.7.17) and sorted with samtools (v1.9).

Q14: Line 558 The threshold value of 10% is too low, which might cause too many false data. Please consider the threshold value of vOTUs clustering (85%).

R14: We thank the reviewer for this suggestion. However, the sequencing depths of CD viral contigs is not high, and for this reason we would like to be as stringent as possible, and avoid false-positive outcomes. Our viral contigs could only recruit every few viral reads from the bulk metagenomes. Using higher thresholds, as suggested by the reviewer, would calculate most viral contigs as zero coverages creating false-negative results due to the low sequencing depths. Below you can find three figures that we generated for this reviewer, using the threshold that he/she suggested (85%) (Fig. a), a 50% threshold (Fig. b) and the 10% threshold (Fig. c) which is used in this study. Considering that almost all (1,622/1,628) of our viral contigs are > 10kb, a threshold of 10%, will result in many/most viral contig to still have > 1kb region covered by reads (identity > 95%). This reduces false-positive outcomes. Roux et al. 2017, reports that increasing thresholds (read mapping identity percentage and length of contig covered), progressively decreases the sensitivity of the analysis, and the false discovery rate (defined as percentage of contigs recovered, that were not part of the initial community). This is something that we would like to avoid with our CD data. However, we report both alpha and beta diversities (plus other indices Chao, Shannon etc), so even with low coverage (viral contigs > 10% of their length covered by metagenome reads) we can still characterize fairly well the viral diversity of CD. We know that alpha diversity can be highly variable when samples are significantly under-sequenced, but beta diversity trends can be recovered even when sequencing depth is highly variable (Roux et al. 2017).

Roux, S., Emerson, J.B., Eloe-Fadrosh, E.A. & Sullivan, M.B. 2017. Benchmarking viromics: An in silico evaluation of metagenome-enabled estimates of viral community composition and diversity. PeerJ. 5:e3817.

R15: Line 544 check grammar.

Many commas were missing in the manuscript, such as Line 129, 192, 222, 410.

Q15: We added commas where necessary, following reviewer's suggestion.

Reviewer #2 (Remarks to the Author):

Q1: The authors took on the endeavor to detect, characterize and compare viruses recovered from deep ocean trenches. I appreciated all the efforts to study these under-sampled environments that will advance our understanding of environmental viruses and their ecological functions. Overall, the topic is important, and the results are exciting to read. However, I do have some major concerns.

1) More thoughts are needed in methodology: example 1: the authors applied one unpublished or not peer-reviewed workflow to detect viral contigs. Although the tools mentioned in the workflow are widely used, the cut-offs and ways of sorting the results are encrypted in the workflow; example 2: suspicious methods to screen putative viral contigs such as 'at least 70% of proteins in the contig were assigned as 'hypothetical protein', 'unknown function'; example 3: identify lytic viruses using VIBRANT that can lead to misinterpretation of the results.

2) The authors need to be careful when citing references to support your discussion. Examples: citing soil viruses for supporting low lysogeny; citing thawed permafrost papers for permafrost ecosystem; citing Paez-Espino et al. (2017) for the suspicious method of screening putative viral contigs.

R1: We thank for the reviewer for finding our results exciting to read. We address this reviewer's concerns on our point-by-point responses.

Q2: Line 35 ‘isolated from, ’, position of the comma.

R2: We deleted comma.

Q3: Line 57-61 and line 69-73 please consider separating the long sentence into two.

R3: We rephrased lines 57-61 (now new lines 57-61) following reviewer’s suggestion. Now it reads: “Viruses show high abundances in marine sediments (10^7 - 10^{10} particles g^{-1} of dry sediment). Yet, viral particles bind firmly to sediments due to electrostatic, “van der Waals” and hydrophobic interactions, which complicate their separation and enumeration from the surrounding sediment matrix ²¹”.

We did the same for lines 69-73 (now new lines 69-71). Now it reads: “Among prokaryotes, Thaumarchaeota and other archaeal lineages in deep-sea sediments, are reported to be more susceptible to viral infections compared to bacterial taxa ²⁷” . We have deleted lines 71-72 (see Q4) to avoid redundancy. The notion in lines 71-72 was already addressed in lines 67-69: “The viral shunt in abyssal and hadal realms is estimated to contribute 35% of labile carbon in those habitats and is believed to sustain the sediment microbiota in hadal sediments by providing easily degradable carbon”

Q4: Line 71-72, ‘the fast decomposition of released viruses following prokaryotic infections’. Do you mean ‘the fast decomposition of viruses released by the lysed prokaryotic host cells’? if so, ‘prokaryotic infection’ is misleading. Please consider re-writing it for clarity.

R4: Please see Q3 response.

Q5: I would encourage the authors to enclose the sequencing and assembly statistics (e.g., quality-filtered reads, Numbers of contigs/scaffolds, N50, numbers of reads that contribute to viral contigs etc.) in the supplementary file for studies using metagenomes/metatranscriptomes.

R5: Thanks for the suggestion. Please find new Supplementary Table 3, that now includes the sequencing and assembly statistics.

Q6: Line 121, what is ‘T3L11’? why this was selected? If there is no particular reason, please consider editing it into ‘from one of the sediment cores (T3L11, 10,908 m)’.

R6: T3L11 was the deepest site sampled during our cruise in Challenger Deep. Also, T3L11 is the only sediment core for which we have metatranscriptome data. We corrected the text following the reviewer’s suggestion and now it reads: “We also generated three metatranscriptome libraries from one of the bottom-axis sediment cores (T3L11: 10,908 m; 6-9, 12-15, 18-21 cmbsf) to gain insights into potential viral activities”. Please see lines 120-122.

Q7: It is a bit worrying that the authors used viral detection workflow that is not published or peer reviewed (Marquet, Mike, et al. "What the Phage: A scalable workflow for the identification and analysis of phage sequences." bioRxiv (2020).).

R7: We understand this reviewer’s concern. The pipeline by Marquet et al., combines 12 tools for phage annotation and identification. We used it primarily to check if viral contigs were present in our data set, and if yes, if they had sizes of > 10kb. Considering many/most viral contigs in CD are novel, it would be beneficial to know the performance of different viral prediction tools for these data. However, because this is not a peer-reviewed pipeline, and also had a highly variable prediction

quality, we were also skeptical and wanted to be as stringent as possible. We therefore performed rigorous manual curation of the data as we explain in lines 129-136. This removed more than 80% of the generated viral contigs from further downstream analysis. The predicted viral contigs used for this study are ~16% (1628/9889) of those initially identified by the pipeline and are retained using consensus metrics of > 95% identity and > 85% coverage.

Following this reviewers' suggestion on Q20, we also now include Supplementary Table 4 that summarizes the screening criteria for identifying the viral contigs, the cut-offs used for each tool and we include the term "putative" viral contigs to be more careful how we refer to them in our study.

We believe that what we report are reliable and uncontaminated data, and that in our effort to be as cautious as possible we have potentially excluded many real viral contigs in the course of our analyses.

Q8: Line 131-134, a total of 1628 contigs were detected and they were clustered in 1628 vOTUs with sequences longer than 10 kb and another six vOTUs with sequences shorter 10 kb? These sentences are confusing.

R8: We apologize for that. We rephrased to avoid any misunderstanding.

Now it reads: "Overall, 1,622/1,628 vOTUs were > 10 kb while six had sizes less than 10 kb (Fig. 2a and Supplementary Table 5)". See lines 134-136.

Q9: Line 135, 'due to the removal of host regions from the proviruses'

R9: Please see our R8 response. We only identify viruses from contigs > 10kb. After viral identifications, we used checkv to estimate viral completeness, which will also remove host regions from proviruses.

Q10: Line 137, what does it mean, 'at least for vOTUs with closely related reference genomes'? if certain criteria were applied to assess the vOTUs using checkV, please write in a full sentence. If not, please consider removing it for clarity.

R10: We apologize for this vague statement. We rephrased accordingly and now it reads: "The degree of completeness and contamination of the CD vOTUs was estimated by comparing the sequences using CheckV⁴⁴ against a large database of environmentally-diverse, and complete viral genomes.". See lines 136-138.

Q11: Line 154, the permafrost samples in Emerson et al. (2018) were classified as 'thawed permafrost'.

R11: We rephrased for accuracy. "wetland sediments⁴⁶, and thawed permafrost⁴⁷ (Fig. 2f)". Line 153.

Q12: Line 164, 'estimated relative abundances'? because the reads coverage was calculated by the reads that were mapped to viral contigs relative to the ones that were not. Reads coverage is just an estimate.

R12: We agree with this reviewer and we corrected this as suggested. See line 168: "The estimated abundances of vOTUs"

Q13: Line 169, do we know the taxonomy of ‘T1L10_NODE_10823’? any close relatives in the reference database?

R13: This would be ideal but unfortunately, we do not know the taxonomy of T1L10_NODE_10823. Also, T1L10_NODE_10823 had no close relatives in the reference databases, and was not affiliated with the identified Thaumarchaeota viruses.

Q14: Line 180, it is worrying to classify the viral contigs that were not detected as ‘prophages’ or ‘potential temperate viruses’ are lytic viruses, although the authors acknowledged the risks of overestimation.

R14: We understand this concern and for this reason we rephrased all text between lines 187-198 and we replotted Figure 2b (now Figure 2e). This figure now describes viral contigs as “Lysogenic” and “Unassigned” (see below). Also, the rephrased text now includes the suggestions of Q15-Q16 by this reviewer.

Please see lines (187-198): “Our results indicated that 1,541 viral contigs (95%) in CD viromes were not assigned to either a lytic or lysogenic lifestyle (Fig. 2e, “undetermined”). It is possible that many/most of these “undetermined” viral contigs belong to viruses that have a lytic lifestyle in hadal depths. This would be consistent with studies of viral communities from surficial sediments collected in different deep-sea oceanic settings (Arctic, Atlantic, Pacific Oceans and Mediterranean Sea; > 1,000 m water depth) that report high viral lysis rates ²⁷. With regard to lysogeny, it was predicted only in 5% of the CD sediment viromes. This differs from deep-sea sediments that showed lysogeny as a more common potential viral lifestyle (e.g. Baltic Sea; ~19% on average) ²⁵ but is more in line with the prediction results that we obtained for deep-sea cold seep sediments (7%)¹⁵ and ocean seawater viromes (3%) ⁷ using VIBRANT ⁴⁹. Nonetheless, our arguments need to be interpreted with caution considering that 95% of viral contigs were not assigned as lytic or lysogenic.

Now, we only predict lysogenic viruses and we re-made figure 2e to reflect this change. All other viruses are classified as “undetermined.”

Fig 2e

Q15: Line 189, please double check the two soil references to support the 5% of lysogeny rate in soil. In soil, the rate is relatively more accurately estimated using induction assay. Lysogeny is quite prevalent in soil. Soil may be not a comparable environment in this case. (ref: Incidence of lysogeny within temperate and extreme soil environments; Prevalence of Lysogeny among Soil Bacteria and Presence of 16S rRNA and trzN Genes in Viral-Community DNA).

R15: We appreciate this important comment from this reviewer. We have now removed soil samples and focused our comparisons only on viral data reported from oceanic settings (e.g., water column, deep-sea cold seep sediments) Please see our R14 for revised text.

Q16: Line 189, could you show me the content in reference 7 indicate the lysogeny percentage of 3%?

R16: We apologize for this misunderstanding. Reference 7 cites the data (and not the 3%) that we used to perform the same prediction analysis of viral lifestyle that we performed for our vOTUs, and we compare the findings. Please see our R14 for revised text.

Q17: Line 210, please use 'viral' instead, as 'virome' refers to the viral fraction that are experimentally enriched from the environmental samples.

R17: We agreed to "viral" following the reviewer's suggestion. Now it reads: "distinct viral components" (line 218).

Q18: Line 255-263, I would suggest shortening this discussion since there are lots of speculations based on unconfident analysis of lytic viruses.

R18: We agree with this reviewer and we shortened this part. The new text on lines 261-266 now reads: "Based on our analyses, lysogeny is a less likely lifestyle (5% assigned) in our identified CD viral contigs. Yet, the inability to assign lifestyle to the majority of the viral contigs (95%) might underestimate the importance of lysogeny, while at the same time preventing us from predicting the lytic viruses in CD. We suggest that lytic infections (if occurring) might be important and affect available nutrient pools across the V-shaped Challenger Deep (bottom-axis vs. slopes sites)."

Q19: Line 321-324, how the modeled protein structure can predict the potential enzymatic activity? How the structure is similar to the reference with validated activity? can some of the residues be aligned to the active sites of these reference structures if any?

R19: All these are great questions. We have now added text in the Methods regarding the modelling of the protein structure that we report on our discussion of putative AMGs (lines 536-543, see below). We also clarify in the discussion how the structure predictions can provide information on the potential enzymatic role that we believe might be useful for the reader (lines 328-339 and below). Overall, in our study we used the web-based Phyre2 services for protein structure prediction (Kelley et al., 2015 *Nature Protocols*; <https://doi.org/10.1038/nprot.2015.053>). The pipeline of Phyre2 uses 4 different algorithms (4 Stages) to provide information on protein structure, function and folding. Structural homologies are estimated against publicly available proteins whose function is known

and structure has been solved using crystallography or other appropriate techniques. We could have performed additional analysis e.g., PROSITE-ExPasy, to describe in more depth the active domains and functional sites at the residue level in our predicted CysC proteins, however we believe that the results we report with Phyre2 are sufficient for the purpose of this study.

Below we include Figure 1 from Kelley et al., 2015 and the Figure legend that describes the pipeline of Phyre2.

For this reviewer our added discussion on lines 328-339:

“The distinct phylogenetic results and the moderate similarity of the CD Cys proteins to those that are publicly available, prompted us to perform protein structure prediction for the CysC protein from the viral contig T1B8_NODE_1222 (Fig. 6c). We used the web-based Phyre2 tool that predicts protein structure and function using homology with known proteins available in protein data banks⁵⁹ (see Methods). The Phyre2 results predicted that CD CysC sequences belong to P-loop containing nucleoside triphosphate hydrolases and specifically to those hydrolases with a structural domain for adenosine-5'phosphosulfate kinase (APS kinase). The top three Phyre2 hits showed that 92-99% of the CysC protein sequences (109-135 residues) have been modelled with 99% confidence and exhibit structural homology with prokaryotic APS kinases. This suggests that CD CysC proteins could catalyze the phosphorylation of APS to 3'-phospho-APS (PAPS), an intermediate step in sulfate assimilation.”

Also, the added text in the Methods on lines 536-543 now reads: “The structure prediction for CysC protein was performed with the web-based Phyre2 tool⁵⁹. Structural homologies were analyzed using models generated by Phyre2 using a confidence threshold of > 98%, and identity threshold of > 29%. The accuracy of the models constructed using Phyre2 is described as extremely high when the sequence identity is above 30-40%. However, lower sequence identities can be equally accurate and useful as long as the confidence threshold is high, which was the case in our examined CysC proteins. The functional domain for CysC was identified and annotated by SMART¹¹³.

Figure 1 (from Kelly et al., 2015). Stage numbers are shown in circles, and elements within a stage are surrounded by a dashed box. Stage 1 (gathering homologous sequences): a query sequence is scanned against the specially curated nr20 (no sequences with >20% mutual sequence identity) protein sequence database with HHblits. The resulting multiple-sequence alignment is used to predict secondary structure with PSIPRED and both the alignment and secondary structure prediction combined into a query hidden Markov model. Stage 2 (fold library scanning): this is scanned against a database of HMMs of proteins of known structure. The top-scoring alignments from this search are used to construct crude backbone-only models. Stage 3 (loop modeling): indels in these models are corrected by loop modeling. Stage 4 (side-chain placement): amino acid side chains are added to generate the final Phyre2 model.

Q20: Line 541-455, how the viral contigs were identified by WtP? Putative viral contigs if detected by more than one tool? Please specify the screening criteria and cut-offs used for each tool.

R20: Please see **R7** and we now provide Supplementary Table 4 that summarizes screening criteria for identifying the viral contigs, the cut-offs used for each tool.

Q21: Line 461-462: ‘at least 70% of proteins in the contig were assigned as ‘hypothetical protein’, ‘unknown function’ was used for screening viruses? This approach is suspicious. Here the authors cited ‘Nontargeted virus sequence discovery pipeline and virus clustering for metagenomic data’. As far as know, Paez-Espino et al. did not use similar approach. More explanation is needed.

R21: Thank you for this question. We will elaborate to avoid potential misunderstandings. Lines 458-481 explain the criteria we used for screening viral contigs; however we rephrased this part for clarity (see lines 463-467). Overall, our criteria for identifying the viral contigs were: 1) contig size (>10kb), 2) presence of 2 or more hallmark viral genes in the contig and 3) absence of any

prokaryotic signature in the contig; in case the contig contained prokaryotic-specific genes it was removed from further analysis. We don't use their pipeline (Paez-Espino et al., 2017), and we apologize if this was interpreted this way. We cited Paez-Espino et al., 2017 because we wanted to show that the authors retained putative viral contigs that mostly contained genes of unknown and hypothetical function as long as 1) contigs included hallmark viral genes and 2) had absence of plasmid or microbial signature gene sequences. Also, Paez-Espino et al., (2017) screened all DNA metagenomic contigs that were more than 5 kb in size, while the authors use as a filter (Filter 1 out of 3) the following: “**metagenomic contigs that had at least 5 hits to viral protein families; AND Total number of genes covered with KO terms on the contig $\leq 20\%$. AND Total number of genes covered with Pfams $\leq 40\%$; AND Total number of genes covered with viral protein families $\geq 10\%$.**”). These criteria identified contigs as viral contigs in their study, although they contained high percentage of unknown function genes. Also, we don't find it surprising that contigs contain genes encoding hypothetical proteins or proteins of unknown function since publicly available databases are not enriched with viral data, especially coming from deep-sea and hadal trench habitats. Further, culture-based experiments that could be more precise in assigning functions to proteins exist primarily for fungi and bacteria from Challenger Deep (at least to our knowledge). Besides Paez-Espino et al (2017), we also now cite Gao et al., 2020 (Gao, SM., Schippers, A., Chen, N. et al. Depth-related variability in viral communities in highly stratified sulfidic mine tailings. *Microbiome* 8, 89 (2020); <https://doi.org/10.1186/s40168-020-00848-3>), that also retained viral contigs that had a total number of genes assigned as “unknown” (annotated with eggNOG v5.0.0 database) for $\geq 80\%$ of the total number of genes on viral scaffolds (> 10 kbp in size) or scaffolds were enriched in hypothetical proteins.

Q22: Missing the method section of mapping metatranscriptome to viral contigs.

R22: We have added text in material and methods. Please see lines (589-592):

“The metatranscriptome reads were also mapped to viral contigs/genes using BWA (v 0.7.17) ¹¹⁷ and then counted reads with aligned length ≥ 50 bp and identity $\geq 95\%$ by CoverM v0.6.1 (parameters: --min-read-percent-identity 0.95 --min-read-aligned-length 50) to generate abundance profile.”

Reviewer #3 (Remarks to the Author):

The manuscript “Ecogenomics reveals novel viromes in the deepest ocean trench on Earth” describes a metagenomic survey of viral populations in the hadal region of the Marianas trench. The authors describe the analysis of metagenomes to identify viral OTUs, analyze these for species distribution, predicted hosts, and potential auxiliary metabolic genes. Though the study presents interesting data about the viral populations of the deepest regions of ocean floor it is very descriptive, with not a lot of evidence supporting the conclusions beyond the observations described. The paper could be improved by addressing the following comments.

Q1: The term ‘ecogenomics’ is used in the title but never defined or discussed in the text of the paper.

R1: The term “ecogenomics” appears routinely now in the literature, and in common-use dictionaries. We do agree with this reviewer that uncommon words in titles or abstracts should

require explanation in the text, but in this case, we feel none is required.

Q2: The first use of the term ‘vOTU’ (line 107) isn’t preceded by a definition of what that means.

R2: Following this reviewer’s suggestion, we now introduce the vOTU definition earlier in text. Now it reads “1,628 virus operational taxonomic units (vOTUs) within the 37 metagenomes and examined their taxonomy, viral community structure, and linkages to prokaryotic hosts.” (See line 105-106)

Q3. In the Results section (lines 131-136) the curation process yields 1628 contigs, but the authors then state that these are further separated into 1628 vOTUs >10kb and 6 < 10kb (i.e. more than the number of contigs). This should be clarified.

R3: We rephrased to avoid any misunderstanding.

Now it reads (See lines 134-136): “Overall, 1,622/1,628 vOTUs were > 10 kb while six had sizes less than 10 kb (Fig. 2d and Supplementary Table 5)”.

Q4. Figure 2a is very confusing and needs (at least) to be better described and have better labels. It’s not clear what the X axis represents (length of contig maybe?). It seems that the dots and lines under the histogram might represent overlap of different methods – though it’s not clear how that’s being represented or what it means.

R4: We apologize for this confusion. We relabeled the entire Fig 2 to clarify.

Also, we rephrased its legend and it now reads: “Figure 2. Overview of CD viromes and bona fide viral contigs (vOTUs) identified in this study. UpSet plot showing (a) the vOTUs predicted from CD metagenomes by the 11 viral predication tools, (b) the different combinations of multiple tools that predicted vOTUs (dot matrix), (c) the number of contigs identified by each tool combination, (d) and the length distribution of all identified viral contigs. Dashed lines indicate the shortest and longest viral contigs, respectively. (e) Bar charts showing the quality and taxonomy of CD viral contigs. (f) Venn diagram of shared viral clusters (genus level) among the five data sets of environmental viruses from CD sediments (this study), other hadal and non-hadal deep-sea sediments (sediments from seven cold seeps¹⁵ and three hadal trenches³⁰), wetland⁴⁶, thawed permafrost⁴⁷ and pelagic sea water (Global Ocean Viromes 2.0)⁷.”

Q5. Reference to Figure 1 in line 149 is a bit confusing – I guess it’s there because the different habitats have just been described?

R5: This is correct. Figure 1 shows our sampling site, as well as those sites that we used data from for comparison.

Q6: Better labeling of the X axis on Figure 3 would be helpful – it’s very hard to interpret as is without reading each of the sample labels. Bars or other graphic indicating groups of samples (trough vs. slope, e.g.) would be useful.

R6: Thank you for this comment. This figure already includes the grouping (slope vs. bottom-axis) that this reviewer requested (please see below). We applied bold font in “Slope” and “Bottom-axis” to make them more distinct.

Q7: Lines 200-202: it's not clear how the distribution of viral populations is consistent with distributions of prokaryotic communities. Needs clarification.

R7: Thank you for pointing this out. We rephrased this and it now reads (lines 206-211): “The distribution of the dominant viral populations, at species level, was also different between the slope and bottom-axis sites (Supplementary Fig. 3). This can be attributed to differences in the geographical isolation and nutrient availability between slope and bottom-axis sites that have been suggested to affect the distribution of prokaryotic communities across the V-shaped CD trench^{38, 39}”

Q8: Figure 4b should have a visual key in the figure to help readers remember the meaning of blue and red bars.

R8: We have added information to the legend to clarify blue (bottom-axis sites) and red (slope sites) bars.

Figure 4 b-f

Q9: Line 220-221. “These prokaryotic MAGs were recovered largely... from the same metagenomes as the viral contigs.” Is confusing – it implies that there are MAGs that were from sources other than the same metagenomes (which I assume there were not)

R9: This reviewer is correct. They are the same MAGs. We rephrased for clarity and now it read (line 229): “These prokaryotic MAGs were recovered from the same metagenomes as the viral contigs.”

Q10: Lines 277-278: “most CD viral populations target” – this is way overstating things given that the number of vOTUs that had predicted hosts was very low. Same with next sentence too-

R10: We agree with the reviewer and we rephrased accordingly. Now it reads on lines 280-283: “The predicted potential prokaryotic hosts for the 14 vOTUs may suggest that CD viruses target specific prokaryotic hosts in these CD sediments; however, this requires cautious interpretation considering that our host predictions were successful for ~1% of the viral population that we identified.”

Q11: Lines 320- . The structural modeling comes out of nowhere - it makes sense, it’s just not adequately described in the results or methods. Nor is the conclusion that the CD CysC can carry out the function (is this the normal function of CysC? What more did predicted structure show?)

R11: This reviewer is right. We have now added text in materials and methods that describes the structural modeling that we performed (please see lines 536-542). Yes, this is the normal function of CysC. It participates in the second step of sulfate assimilation, and is a conserved protein. The whole pathway of sulfate assimilation primarily serves to provide available sulfur (in the form of sulfite) that can be used for the synthesis of S-bearing amino acids (cysteine and methionine). AMGs encoding proteins involved in the pathway of sulfate (or sulfur) assimilation (e.g., cysC, PAPS) have been found to be carried by viruses in other oceanic settings which are distinct from the Challenger Deep (e.g., Cariaco Basin; Mara et al., 2020), and have been identified from publicly available metagenomic and metatranscriptomic data (e.g. from hydrothermal environments, freshwater lakes, and Tara Ocean; Kieft et al 2021).

Q12: Lines 332: how is the statement “our data indicate accumulation of heavy metals” supported? Was this from other measurements taken of the sediments? If so this needs to be described more fully here.

R12: Thank you for this comment. Yes, we have data on heavy and trace metals from the CD. Along with sediments for DNA and RNA extractions we collected sediments for arsenic, mercury and selenium analysis. The arsenic/mercury/selenium data are already published (Zhou et al., 2022) and we also plotted Supp Figure 8 to show this.

Zhou Y, Mara P, Cui G, et al. Microbiomes in the Challenger Deep slope and bottom-axis sediments[J]. Nature Communications, 2022, 13(1): 1-13.

Q13: Line 338-339: “increase the viral fitness towards the potential toxic effect of the arsenic accumulation.” Doesn’t make sense.

R13: We rephrased and now reads (lines 352-353): “could enhance the heavy metal detoxification mechanisms of the prokaryotic hosts, and thus, increase the viral fitness”.

Q14: A comparison of AMGs from another set of metagenomes would be helpful: are these interesting observations or just what’s seen everywhere?

R14: We agree with this reviewer. However, CD AMGs were manually curated which makes it difficult to manually check and compare all of them with AMGs from other datasets. For this reason, we chose two AMGs related to sulfur metabolism (*cysC* and *cysH*), to perform similarity and phylogenetic analyses. We report our findings in lines 315-326, Figure 6b, Supplementary figure 7, Supplementary figure 8 and Supplementary Table 12. Overall, these CD AMGs were more similar to their closest homologues identified from environmental viromes, and less similar to the proteins deposited in the eggNOG database. As mentioned in our response in Q11 some of the AMGs have been reported in other ecosystems which are distinct from the Challenger Deep.

Lines 315-326:

“To understand the origin of the putative CD AMGs related to sulfur assimilation, we recruited the top five a) CysC proteins from the eggNOG database (v5.0) with close homology to our CD viral CysC proteins, and b) CysC-encoding AMGs predicted from different viromes ^{7, 15, 46, 48, 57}, respectively. The similarity between our CD CysC-encoding AMGs and those CysC proteins deposited in the eggNOG database (v5.0) ranged from 27% to 47% (Supplementary Table 12). These similarity percentages were lower when we compared our putative CysC-encoding AMGs, with those identified in global-scale ocean virome datasets, including those from deep-sea sediments and permanently anoxic basins ^{7, 15, 46, 48, 57} (34% to 61%; Supplementary Fig. 7a). The phylogenetic analysis for three of our CD CysC proteins showed that they are distinct from their prokaryotic CysC homologs but cluster with CysC proteins from the different viromes referred to above (Fig. 6b).”

Figure 6b Maximum-likelihood phylogenetic tree of CD CysC proteins. The CysC proteins predicted in CD viral genomes were used to construct a phylogenetic tree using homologous CysC proteins deposited in the eggNOG database (V5.0) and publicly available viromes. CysH proteins were used as an outgroup. We also included two CysC homologs from the Uniport database (in blue) with experimental evidence of function at the protein level. Bootstrap values (1,000 replicates) $\geq 70\%$ are indicated at nodes.

Supplementary Figure 7. Similarity of each Cys protein to the five closest homologues in public viromes and the eggNOG database, respectively. Three cysC (a) and six cysH proteins (b) from CD viral contigs.

The viromes from the following references were used to pick viral AMGs related to sulfur assimilation (cysC and cysH):

- Dalcin Martins P, *et al.* Viral and metabolic controls on high rates of microbial sulfur and carbon cycling in wetland ecosystems. *Microbiome* **6**, 138 (2018).
- Mara P, *et al.* Viral elements and their potential influence on microbial processes along the permanently stratified Cariaco Basin redoxcline. *ISME J* **14**, 3079-3092 (2020).
- Gregory AC, *et al.* Marine DNA viral macro- and microdiversity from pole to pole. *Cell* **177**, 1109–1123 (2019).
- Li Z, *et al.* Deep sea sediments associated with cold seeps are a subsurface reservoir of viral diversity. *ISME J* **15**, 2366–2378 (2021).
- Zhao J, *et al.* Novel viral communities potentially assisting in carbon, nitrogen, and sulfur metabolism in the upper slope sediments of Mariana Trench. *mSystems* **7**, e01358-01321 (2022).
- Jian H, *et al.* Diversity and distribution of viruses inhabiting the deepest ocean on Earth. *ISME J*, (2021).

Reviewers' comments:

Reviewer #1 (Remarks to the Author):

The authors address most of my points. However, they only revised and added a few sentences for my main concern. Considering the recent paper of Zhao et al., 2022, the authors should discuss the present data together with Zhao's data. For example, what is the ecological and biogeochemical significance that very distinct viral population were observed among upper slope, lower slope and bottom axis of Challenger Deep?

Reviewer #2 (Remarks to the Author):

The authors addressed most of my comments. I have additional comments as follows:

1. Line 42: 'contribute in understanding' reads awkward. 'Contribute to understanding'.
2. Mixed use of virome and metagenome. As mentioned in the previous comment, virome is metagenome of the viral fraction that is experimentally enriched from the samples. (ref: 'Viromes outperform total metagenomes in revealing the spatiotemporal patterns of agricultural soil viral communities'). If the authors do not want to follow the definition that is widely accepted by the community, please define virome in front and use it consistently throughout the manuscript.
3. Line 63, missing 'e.g.,' after 'sediment composition'? Additionally, pH and size are not 'sediment composition' and are generally described as physiochemical factors or properties.
4. Missing references for 'encode putative auxiliary metabolic genes (AMGs) involved in carbon and sulfur metabolisms' in line 74. If it shares the same references with line 73, please consider moving the reference index to the end of the sentence.
5. It is not common to have a Supplementary table in Introduction.
6. It is weird to have 'Finally' in line 143.
7. Is the clustering of vOTUs consistent with that of vConTACT clusters for the CD viral sequences?
8. Line 162, vConTACT2, not 'vcontact2'.
9. Line 166, please note that 'Siphoviridae, Myoviridae and Podoviridae' families are abolished in the latest release of ICTV. Please double check this and update it accordingly.

Reviewer #3 (Remarks to the Author):

The authors have addressed my concerns - I see no further issues.

Response to the Reviewers:

We appreciate the reviewers for taking the time to evaluate our manuscript for their positive and constructive feedback.

Reviewer #1 (Remarks to the Author):

Q1. The authors address most of my points. However, they only revised and added a few sentences for my main concern. Considering the recent paper of Zhao et al., 2022, the authors should discuss the present data together with Zhao's data. For example, what is the ecological and biogeochemical significance that very distinct viral population were observed among upper slope, lower slope and bottom axis of Challenger Deep?

R1: We thank the Reviewer's positive assessment of our work and constructive suggestions. We feel we can only devote a few sentences on what this reviewer suggests because both our paper and the Zhao paper present data from one vertical sampling location along Challenger Deep's trench and bottom-axis. Until additional locations are sampled in the future, it is impossible to interpret the variations in communities that we see between Zhao et al.'s sites ranging from 5.4 to 6.7 km water depth vs. our 13 sites (slope and bottom-axis). Differences could be in DNA recovery between the studies, methods for metagenome assembly and extraction of viral data from those, or in situ chemistry variations between samples (that we observed, and reported also at Zhou et al., 2022). A sentence has been added to the text (lines 214-217) that says: "Differences in viral communities at discrete depths observed in this study, and between this study and upper CD slope ⁴⁸ may possibly reflect in situ variations in available nutrients, and/or variations in DNA recovery or methods used for metagenome assembly and extraction of viral data."

Reviewer #2 (Remarks to the Author):

The authors addressed most of my comments. I have additional comments as follows:

Q1. Line 42: 'contribute in understanding' reads awkward. 'Contribute to understanding'.

R1: Thank you. We rephrased as reviewer suggested.

Q2. Mixed use of virome and metagenome. As mentioned in the previous comment, virome is metagenome of the viral fraction that is experimentally enriched from the samples. (ref: 'Viromes outperform total metagenomes in revealing the spatiotemporal patterns of agricultural soil viral communities'). If the authors do not want to follow the definition that is widely accepted by the community, please define virome Infront and use it consistently throughout the manuscript.

R2: We agree on following the widely accepted definition. We replaced "viromes" with "viral sequences", "viral communities", "viral datasets" and "viruses" where necessary (e.g., lines 35, 72, 78, 123, 147, 157, 158, 190, 196, 199, 202, 220, 252, 293, 312, 322, 326, 330, 366, 375, 378, 461, 462, 504, 531, 533), including also the title of the manuscript.

Q3. Line 63, missing 'e.g.,' after 'sediment composition'? Additionally, pH and size are not 'sediment composition' and are generally described as physiochemical factors or

properties.

R3: We rephrased. Now it reads: “The challenges of efficiently separating viral particles from the sediments are due to the features of the virus (e.g., size, isoelectric point) and the sediment physiochemical properties (e.g., size, mineralogy, pH) that control the type and strength of interactions between viral and sediment particles.”

Q4. Missing references for ‘encode putative auxiliary metabolic genes (AMGs) involved in carbon and sulfur metabolisms’ in line 74. If it shares the same references with line 73, please consider moving the reference index to the end of the sentence.

R4: That is correct. Now reference is cited at the end of the sentence in line 74.

Q5. It is not common to have a Supplementary table in Introduction.

R5: We agree. Supplementary table 1 is now deleted.

Q6. It is weird to have ‘Finally’ in line 143.

R6: We rephrased and now reads: “We also identified that 19% of vOTUs (316/1,628) had at least 20% of genes mapped by > 1 metatranscriptomic reads in our bottom-axis metatranscriptomic libraries. See lines 142-144.

Q7. Is the clustering of vOTUs consistent with that of vConTACT clusters for the CD viral sequences?

R7: Yes, it is. We chose to present vOTUs at species level and the vConTACT2 clusters for genus level.

Q8. Line 162, vConTACT2, not ‘vcontact2’.

R8: Thank you for pointing this out. Now have now corrected to “vConTACT2”.

Q9. Line 166, please note that ‘Siphoviridae, Myoviridae and Podoviridae’ families are abolished in the latest release of ICTV. Please double check this and update it accordingly.

R9: The viral reference sequences in NCBI (RefSeq-release208 downloaded on January 4, 2022) that were used in this study to annotate the CD viruses, utilized the old/commonly described nomenclature for Myo-, Podo-, and Sipho- viridae. We add the following statements on our manuscript to clarify. See lines167-172 and 493-495:

Lines167-172: We note that since the time of data freeze for preparation of this manuscript, the taxonomy of phages has undergone a revision described in Walker et al. 2021⁴⁹, and is now implemented by the International Committee on Taxonomy of Viruses (ICTV). As a result, the taxon naming will need to be updated by interested users of our data with the new taxon names that were approved after our analyses were completed.

Lines 493-495: In brief, we used blastp (version 2.9.0+) to query all proteins from CD vOTUs against NCBI viral RefSeq database release 208 (<https://ftp.ncbi.nlm.nih.gov/refseq/release/viral/>, downloaded on January 4, 2022).

Walker, P.J., Siddell, S.G., Lefkowitz, E.J. et al. Changes to virus taxonomy and to the International Code of Virus Classification and Nomenclature ratified by the International Committee on Taxonomy of Viruses (2021). Arch Virol 166, 2633–2648 (2021).

Reviewer #3 (Remarks to the Author):

Q1. The authors have addressed my concerns - I see no further issues.

R1: We thank this reviewer for endorsing our work for publication.